# Role of phosphorus in the seasonal deoxygenation of the East China Sea shelf

Arnaud Laurent[1], Haiyan Zhang[1,2], Katja Fennel[1]

[1] Department of Oceanography, Dalhousie University, Halifax, Nova Scotia, Canada
[2] School of Marine Science and Technology, Tianjin University, Tianjin, China

*Correspondence to*: Arnaud Laurent (arnaud.laurent@dal.ca)

**Abstract.** The Changjiang is the largest river in Asia and the main terrestrial source of freshwater and nutrients to the East China Sea (ECS). Nutrient concentrations have long been increasing in the Changjiang, especially after 1960 with urbanization, the development of industrial animal production, and fertilizer application in agriculture, resulting in coastal eutrophication and recurring summer hypoxia. The supply of anthropogenic nitrogen (N) exceeds that of phosphorus (P) relative to the Redfield ratio. This results in seasonal P limitation in the Changjiang plume. P limitation and its effects on primary production, respiration and hypoxia in the ECS have not been studied systematically yet although such knowledge is needed to understand bloom dynamics in the region, to assess the consequences of altered nutrient loads, and to implement nutrient reduction strategies that mitigate hypoxia. Using a coupled physical-biogeochemical model of the ECS that was run with and without P limitation, we quantify the distribution and effects of P limitation. The model shows that P limitation develops eastward of the Changjiang Estuary and on the Yangtze Bank but rarely southward along the Zhejiang coast. P limitation modifies oxygen sinks over a large area of the shelf by partly relocating primary production and respiration offshore, away from the locations prone to hypoxia near the Changjiang Estuary. This relocation drastically reduces sediment oxygen consumption nearshore and dilutes the riverine-driven primary production and respiration over a large area offshore. Our results suggest that the hypoxic zone would be 48% larger in its horizontal extent, on average, if P limitation was not occurring. Results are summarized in a conceptual model of P limitation on the ECS shelf that is also applicable to other systems. Then we carried out nutrient reduction simulations which indicate that, despite the effect of P limitation on hypoxia, reducing only P inputs as a nutrient reduction strategy would not be effective. A dual N+P nutrient reduction strategy would best mitigate hypoxia. The model results suggest that decreasing the

size of the hypoxic zone by 50% and 80% would require reductions in N+P load of 28% and 44%, respectively.

## 1    Introduction

The expansion of eutrophication-driven hypoxic conditions, i.e. $O_2 < 2$ mg l$^{-1}$ or 62.5 mmol m$^{-3}$, has been a growing issue for the coastal ocean in the past decades (Diaz and Rosenberg, 2008; Fennel and Testa, 2019). Coastal eutrophication occurs when excess nitrogen (N) and phosphorus (P) nutrients, associated mainly with the use of synthetic fertilizer in agriculture, wastewater discharge in urban areas, and livestock farming (Alexander et al., 2008; Chen et al., 2020) reach the coastal ocean where they stimulate primary production, the development of harmful algal blooms (Glibert et al., 2014; Heisler et al., 2008), and subsequently respiration and $O_2$ depletion in subsurface waters (Fennel and Testa, 2019). Low $O_2$ can have detrimental effects on the benthos (Levin et al., 2009), the foodweb (Kidwell et al., 2009), and ultimately economic consequences for fisheries (Rabalais and Turner, 2001; Smith et al., 2017).

In large eutrophicated estuaries and river plumes (e.g., Mississippi River plume, Changjiang Estuary, Chesapeake Bay) phytoplankton growth is typically limited by light in winter as well as in the highly turbid waters in the vicinity of the river outflow, whereas maximum production often occurs downstream where both light and nutrients are plentiful (Cloern, 1987; Fennel et al., 2011; Lohrenz et al., 1997, 1999). In downstream waters, productivity is controlled by the availability of nutrients (Fennel and Laurent, 2018; Malone et al., 1996).  There, the stoichiometry of dissolved inorganic N (DIN), P (DIP), and possibly silicate (Si) in the river source will partly determine where, when and which nutrient is most limiting (Dagg et al., 2004; Laurent et al., 2012).

The stoichiometry of dissolved inorganic N (DIN) and P (DIP) inputs from rivers has been modified by the onset of cultural eutrophication. Sources of DIN are usually spatially diffuse, associated with the use of fertilizer for agriculture (Seitzinger and Harrison, 2008). Although diffuse sources of DIP are also important in intensive farming regions (Boesch, 2019; Wurtsbaugh et al., 2019), sewage point sources can dominate DIP export (Harrison et al., 2010). The construction of dams in a river basin may also alter the delivery of particulate phosphorus to the coastal ocean (Beusen et al., 2016; Hu et al., 2020). Successful point source mitigation has modified nutrient stoichiometry in rivers resulting in a growing

excess of N and DIN:DIP ratios significantly larger than the Redfield ratio of 16 (Westphal et al. 2020). This leads to P-limitation of phytoplankton growth in coastal waters and the modification of bloom dynamics (Laurent et al., 2012) and consequently hypoxia (Laurent and Fennel, 2014, 2017).

One of the world's largest eutrophication-induced hypoxic zones is in the East China Sea (ECS), a large and highly productive marginal sea on the western edge of the North Pacific surrounded by Taiwan, China, Korea, and Japan. The main terrestrial source of freshwater and nutrients to the ECS is the Changjiang (Tong et al., 2015), the largest river in Asia (Liu et al., 2003). Nutrient concentrations have long been increasing in the Changjiang, especially after 1960 with the development of urbanization, industrial animal production, and fertilizer application in agriculture (Dai et al., 2011; Jiang et al., 2014; Liu et al., 2009, 2018; Strokal et al., 2016; Wang et al., 2020; Yan et al., 2003, 2010). Coastal eutrophication within the Changjiang plume leads to recurrent algal blooms (Chen et al., 2019a; Li et al., 2014; Wang and Wu, 2009) and the development of subsurface hypoxic conditions in summer (Li et al., 2011; Ning et al., 2011; Wang et al., 2016). Although freshwater (stratification) and wind (destratification) are important for the establishment and sustainment of hypoxic conditions (Zhang et al., 2020), the Changjiang's N load has been shown to be the main contributor to hypoxia on the shelf (Große et al., 2020).

Typically, primary production is limited by light in the Changjiang estuarine and sediment plume waters, whereas nutrient limitation may occur in the transition zone and further offshore (Harrison et al., 1990; Li et al., 2021; Zhu et al., 2009). As elsewhere, the unbalanced supply of DIN and DIP from the Changjiang (Liu et al., 2003, 2018; Shi et al., 2022) led to the development of P limitation on the shelf (Li et al., 2009; Wang et al., 2015). Based on laboratory studies and local observations, there are indications that high N:P and the onset of P limitation on the ECS shelf alter the species composition of phytoplankton blooms (Li et al., 2009; Ou et al., 2020; Xing et al., 2016; Zhou et al., 2008). Despite this knowledge, P limitation has not been studied systematically in the region and its effects on primary production, respiration and hypoxia remain unknown. However, understanding the effects of nutrient limitation on $O_2$ biogeochemistry is essential for assessing the consequences of altered nutrient loads on hypoxia dynamics and therefore to implement nutrient reduction strategies to mitigate hypoxia on the shelf. Preliminary N load reduction experiments in the Changjiang show that significant N reduction is

necessary to mitigate hypoxia (Große et al., 2020; Zhou et al., 2017). P was not included in these experiments but need to be considered to assess the efficiency of single versus dual nutrient management strategies.

Here, through simulations with and without P limitation in the coupled circulation-biogeochemical model of Zhang et al. (2020), we investigate nutrient limitation in the ECS. The goal is to quantify the effects of P limitation and provide insights for reducing hypoxia in the region. First, we validate the model with observed nutrient concentrations. We then describe the occurrence and spatial distribution of P limitation on the shelf and its effect on $O_2$ sources and sinks and hypoxia. Using these results, we develop a conceptual model of P limitation for the ECS. Finally, by conducting scenario simulations with reduced nutrient loads from the Changjiang, we investigate the efficiency of different nutrient reduction strategies to mitigate hypoxia on the shelf.

## 2    Methods

### 2.1    Model description

The circulation model is a regional implementation of the Regional Ocean Modelling System (ROMS, version 3.7; Haidvogel et al., 2008) configured for the East China Sea and surrounding areas (Bian et al., 2013; Figure 1). The grid has a 1/12˚ resolution and 30 vertical layers with increased resolution near the surface and bottom. The circulation model uses the recursive Multidimensional Positive Definite Advection Transport Algorithm (MPDATA) for the horizontal advection of tracers and a third-order upwind scheme for the advection of momentum. Vertical mixing is parameterized using the Generic Length Scale (GLS) turbulence closure scheme (Umlauf and Burchard, 2003).

The circulation model is coupled online with the biogeochemical model of Fennel et al. (2006, 2011) that was extended to include phosphate (Laurent et al., 2012), $O_2$ (Fennel et al., 2013), river dissolved organic matter (DOM; Yu et al., 2015) and a light-attenuation scheme that simulates higher attenuation in shallow areas and in the river plume (Zhang et al., 2020). The model has 10 state variables: phytoplankton, chlorophyll, zooplankton, nitrate and ammonium (DIN), phosphate (DIP), $O_2$, and three detritus pools (small and large detritus, river DOM). Silicate (Si) is not included in the model as it is assumed to be non-limiting. During the simulation period, observed Si/N and Si/P were close to and above

Redfield, respectively (Shi et al., 2022). The single phytoplankton group includes all phytoplankton types and therefore does not differentiate between diatoms and dinoflagellates. Zhang et al. (2020) showed that these choices are appropriate to represent the bulk surface chlorophyll in the Changjiang plume, as well as spatial and temporal variations in subsurface $O_2$. At the sediment-water interface, sinking particulate organic N (PON) and P (POP) are instantaneously remineralized into ammonium and phosphate, respectively, accounting for a fraction of PON lost through denitrification (Fennel et al., 2006). Sediment $O_2$ consumption (SOC) is parameterized assuming a linear relationship with denitrification (Fennel et al., 2013; Seitzinger and Giblin, 1996).

The model includes seven rivers. Daily freshwater discharge observations from Datong Hydrological Station (Figure 1) and monthly $NO_3$ and $PO_4$ loads from the Global NEWs model (Wang et al., 2015) are used to represent Changjiang conditions. Monthly or annual climatologies are used for the other rivers. Boundary conditions for temperature, salinity, $NO_3$, $PO_4$ and $O_2$ are specified from the World Ocean Atlas (Garcia et al., 2013a,b; Locarnini et al., 2013; Zweng et al., 2013), whereas horizontal velocities and sea surface elevation use the SODA data set (Carton and Giese, 2008). A small positive value is used for all other biological variables. Surface forcing is prescribed using the ECMWF ERA-Interim dataset (Dee et al., 2011).

A schematic of the biogeochemical model is provided in Figure 1. The model equations are available in the supporting information of Laurent et al. (2017); setup and validation are described in detail in Zhang et al. (2020). Biogeochemical model parameters are also available in Zhang et al. (Table S1).

## 2.2   Nutrient limitation

Phytoplankton growth is limited by light ($E$), temperature ($T$) and the most limiting nutrient (either DIN or DIP) such that the specific phytoplankton growth rate ($\mu$; $d^{-1}$) is calculated as follows:

$$\mu = \mu_{\mathrm{max}}(E, T) \cdot L_{\mathrm{tot}} \tag{1}$$

where $\mu_{\mathrm{max}}(E, T)$ is the light- and temperature-dependent maximum growth rate of phytoplankton and $L_{\mathrm{tot}}$ ($0 < L_{\mathrm{tot}} < 1$) is the minimum of the nutrient limitation factor for DIN ($L_N$) and DIP ($L_P$). $L_{\mathrm{tot}}$ is calculated from nitrate ($NO_3$), ammonium ($NH_4$) and DIP using Equations 2-5 below.

$$L_{\mathrm{NO_3}} = \frac{\mathrm{NO_3}}{k_{\mathrm{NO_3}}+\mathrm{NO_3}} \cdot \frac{1}{1 + \mathrm{NH_4}/k_{\mathrm{NH_4}}} \tag{2}$$

$$L_{\mathrm{NH_4}} = \frac{\mathrm{NH_4}}{k_{\mathrm{NH_4}}+\mathrm{NH_4}} \tag{3}$$

$$L_{\mathrm{N}} = L_{\mathrm{NO_3}} + L_{\mathrm{NH_4}} \tag{4}$$

$$L_{\mathrm{P}} = \frac{\mathrm{DIP}}{k_{\mathrm{DIP}}+\mathrm{DIP}} \tag{5}$$

Following Laurent et al. (2012), we assume in the following analysis that N is limiting when $L_{\mathrm{tot}} < 0.85$ and $L_{\mathrm{N}} < L_{\mathrm{P}}$, whereas P is limiting when $L_{\mathrm{tot}} < 0.85$ and $L_{\mathrm{P}} < L_{\mathrm{N}}$. Although this threshold is somewhat arbitrary, our assumption is only used to describe the regions where N and P are most limiting and therefore the selected value does not influence the simulation results, only their interpretation.

## 2.3   Simulations

All the simulations were run for 8 years with daily output. The first 2 years were considered spin-up and the period 1 January 2008 to 31 December 2013 was used for analysis. The "baseline" simulation, as described above, is identical to Zhang et al. (2020). To assess the effect P limitation, the baseline is compared to a simulation without P limitation ("noPlim"), that is implemented by disabling P in the biological model but otherwise keeping everything identical to the "baseline" simulation. Because this model did not include P, it simulated the situation without any P limitation. In addition, 15 scenario simulations were carried out where riverine N, riverine P and riverine N and P concentrations were decreased stepwise in 20% increments. Otherwise, these simulations were identical to the baseline simulation. Each reduction level is applied to all nutrient subspecies (i.e., dissolved inorganic, dissolved and particulate organic).

For analysis, the shelf region adjacent to the Changjiang Estuary (CE) is divided into 6 zones. Zone 1 represents the Jiangsu coastal area (z<25m, 31.70<lat<33.50). Zones 2 and 3 are the northern and southern hypoxia cores, respectively, defined in Zhang et al. (2020). Zones 4 represents the Yangtse Bank area (z<50m, 30.75<lat<33.50), whereas Zones 5 (30.75<lat<33.50) and 6 (28.52<lat<30.75) represent the northern and southern deep shelf waters (50<z<100m), respectively.

# 3    Results

## 3.1    Riverine input

DIN and DIP concentrations are high year-round in the Changjiang (Figure 2), although DIP concentration tends to peak in the fall, whereas DIN concentrations can vary from year to year. DIN and DIP loads are highly correlated with freshwater discharge (Pearson's correlations of 0.95 and 0.98, respectively) and therefore their annual maxima occur at the time of highest discharge, usually from June to August. The DIN:DIP ratio is 69 on average, well above the Redfield ratio of 16, and ranges from 54 to 78. Hence, nutrient loads from the Changjiang are conducive to P limitation on the continental shelf near the CE.

## 3.2    Surface nutrient concentration

In fall and winter, the Changjiang plume is restricted to the coast and the surface $PO_4$ concentration is high ($>2$ mmol m$^{-3}$) in the CE and Hangzhou Bay (Figure 3a, d). $PO_4$ concentration in the plume is at an annual maximum at this time and $PO_4$ is transported southwestward with the plume along the Zhejiang coast. The circulation reverses in spring when the plume transports Changjiang waters offshore, northeastward over the Yangtze Bank (Figure 3b). Offshore transport is at its annual maximum in summer (Figure 3c). $PO_4$ is ~2 mmol m$^{-3}$ in the CE but plume concentrations decrease to very low values offshore (Figure 3e, g, h, k). $NO_3$ follows similar seasonal and spatial patterns, but plume concentrations remain elevated offshore in summer (20–40 mmol m$^{-3}$, Figure S1), contrasting with $PO_4$ concentrations.

Simulated surface nutrient concentrations are compared with the observations of Gao et al. (2015) in Figure 3e–k ($PO_4$) and Ge et al. (2020) in supporting Figure S1 ($NO_3$). $PO_4$ and $NO_3$ concentrations compare reasonably well with Pearson's correlation coefficients of 0.75 and 0.84, respectively. The model simulates the observed CE-offshore gradient as well as the magnitude of nutrient concentrations in the CE and the Changjiang plume. The model also captures the locations of $PO_4$ and $NO_3$ depletion in offshore waters in summer.

## 3.3 P limitation on the ECS shelf

The spatial distribution of nutrient limitation is shown in Figure 4. As mentioned earlier, we assume hereafter that N or P limitation occur when the limitation factor is < 0.85 and otherwise light-limitation or no limitation. In Figure 4, the daily occurrence of such conditions for N and P is presented as an annual average (days of N or P limitation per year). Near the CE, nutrient concentrations are high (see Figure 3 and S1) and neither N (left panel) nor P (right panel) are limiting. Nonetheless, light can limit phytoplankton growth in this region (Zhang et al., 2020). Offshore ($z > 50$ m) and outside the freshwater plume and along the northern Jiangsu coast, N is limiting for most of the year. P limitation occurs for 2–3 months near and on the Yangtze Bank in zones 2 and 4. It also occurs occasionally further east and in zone 3 but never in the coastal area north of the CE where N is limiting. N limitation is prevalent offshore (zones 5 and 6).

Spatial variability in P limitation occurs mainly along a west–east axis from the CE, in particular along the CE-Jeju Island (JI) line (northeastward, Figure 1). The variability along this axis is presented through CE-JI transects in Figure 5, Figure 6 and supporting Figure S2. P limitation develops about 100 km offshore the CE between May and September (Figure 5) and is found sporadically all the way to Jeju Island in August-September. Outside of a 100-150 km wide area that forms the core of seasonal P limitation, mostly within zone 4, the occurrence of P limitation is variable and depends on the offshore expansion of the Changjiang plume, as represented by the 29 isohaline on Figure 5. For example, in 2010 (high discharge, Figure 2) P limitation occurred over $68.0 \times 10^3$ km$^2$ on average (May–August), extending all the way to Jeju Island for >2 weeks in August. In comparison, P limitation covered only $31.4 \times 10^3$ km$^2$ on average in 2011 (low discharge). The maximum annual extent of P limitation usually occurs in late July–early August ($95.4 \times 10^3$ km$^2$, 2008-2012 average, Table 1). The annually integrated area of P-limitation is highly correlated with freshwater discharge from the Changjiang ($r = 0.79$, $p = 0.06$). Higher discharge promotes a larger, and more sustained area of P-limited surface waters.

In most years, the expansion of P limitation along the transect occurs over two periods, one of limited spatial extent in the spring and the main one in summer (Figure 5). In spring and summer when the plume does not reach Jeju Island, the shelf area surrounding the island is strongly N-limited.

The vertical distribution along the transect line indicates that P limitation is well developed in late July and occurs mainly in the upper 5 m of Zone 4, where the offshore edge of the plume is located (Figure 6a). Further offshore, outside of the plume, strong N limitation occurs within the upper 10 m. P limitation is more variable and starts to break down in late August (Figure 6d).

## 3.4 Consequences of P limitation

The effects of P limitation are quantified from the difference between the baseline and NoPlim simulations. Along the CE-JI transect, P limitation reduces primary production by 20 and 45 mmol $O_2$ m$^{-2}$ d$^{-1}$ on average in July in zones 2 and 4, respectively (Figure 6b, supporting Table S1). The reduction occurs at the surface, whereas an increase is found just below (zone 4) and offshore (+45.0 mmol $O_2$ m$^{-2}$ d$^{-1}$ in zone 5). The decrease in primary production translates into sediment $O_2$ consumption (SOC) which decreases significantly around zone 4 (-32.9 mmol $O_2$ m$^{-2}$ d$^{-1}$, supporting Figure S2) and into water column respiration (WR) that is also reduced at the surface in zone 4 (Figure 6c). This WR reduction is compensated by a subsurface increase thus, integrated over the water column, the change in WR is negligible in zone 4 in July (supporting Table S1). The change in respiration can be explained by the effect of particulate organic matter (POM) concentration on POM flux (supporting Figure S2). As primary production decreases at the surface of zone 4, less organic matter settles down to the bottom (-4.6 mmol N m$^{-2}$ d$^{-1}$). Concurrently, as POM aggregation is proportional to the square of small POM concentration (phytoplankton + small detritus), POM sinking rate decreases and proportionally more remineralization occurs in the water column and less in the sediment. This explains the subsurface increase in water column respiration in zone 4 despite the decrease in surface primary production. Further offshore in zone 5, WR increases as well (+5.20 mmol $O_2$ m$^{-2}$ d$^{-1}$).

When P limitation breaks down in late August (Figure 6d-f), primary production is still reduced near the coast (-25.0 mmol $O_2$ m$^{-2}$ d$^{-1}$ in zone 2) but increases in the upper 5m in the northeastern part of zone 4 and in zone 5 (+17.0 mmol $O_2$ m$^{-2}$ d$^{-1}$). Concomitantly, a significant increase in water column respiration occurs throughout the water column in the northeastern part of zone 4 (+4.90 mmol $O_2$ m$^{-2}$ d$^{-1}$ overall) and in the surface waters of zone 5 (+9.90 mmol $O_2$ m$^{-2}$ d$^{-1}$).

Figure 7 provides the spatially integrated seasonal evolution of the effect of P limitation on production and respiration. Note that the zones have different sizes (Figure 1) and therefore, at the same magnitude, the effect shown in Figure 7 is spatially more concentrated in smaller areas (zone 2) and more diffuse in larger areas (zones 4-5). Spatially averaged values are also available for each zone in supporting Figure S3. In zone 2, P limitation has a negative effect on primary production that is reduced from April to September ($-546 \times 10^8$ mol $O_2$ yr$^{-1}$), whereas it supports primary production in zone 5 from May to October, with a peak in July-August ($+684 \times 10^8$ mol $O_2$ yr$^{-1}$, Figure 7a). In the intermediate zone 4, primary production decreases from April to July and then increases between August and October ($-197 \times 10^8$ mol $O_2$ yr$^{-1}$). This temporal switch is also found when the effect of P limitation on primary production is integrated over the shelf (zones 1-6, grey area). Overall, the integrated change in primary production is negligeable relative to total PP ($-34.0 \times 10^8$ mol $O_2$ yr$^{-1}$, supporting Table S2). The spatial relocation also occurs in the $O_2$ sinks but simultaneously, there is a shift in respiration from the sediment to the water column (Figure 7b-c). Water column respiration does not change significantly in zone 2 ($-21.0 \times 10^8$ mol $O_2$ yr$^{-1}$) but increases in zones 4 (August-October, $+188 \times 10^8$ mol $O_2$ yr$^{-1}$) and 5 (June-October, $+329 \times 10^8$ mol $O_2$ yr$^{-1}$, Figure 7b). Sediment respiration increases simultaneously in zone 5 ($+202 \times 10^8$ mol $O_2$ yr$^{-1}$, Figure 7c), somewhat compensating the decrease in zone 2 ($-271 \times 10^8$ mol $O_2$ yr$^{-1}$). A large decrease in sediment respiration occurs in zone 4 between May and August ($-553 \times 10^8$ mol $O_2$ yr$^{-1}$, Figure 7c). The shift in respiration is clear when integrating over the shelf (shaded areas in Figure 7b-c). The negative effect on sediment respiration is larger than the positive effect on water column respiration and therefore the total change in respiration over the shelf (zones 1-6) is $-164 \times 10^8$ mol $O_2$ yr$^{-1}$ (supporting Table S2).

The effect of P limitation on respiration influences the duration and spatial extent of hypoxia, mainly north of 30°N (Figure 8a). The largest effect occurs in the region adjacent to the CE where hypoxia duration decreases by up to 31 days per year on average. Hypoxia extends further north/northeast into zone 4 without P limitation; hypoxia is less frequent in this area but can last up to 20 days per year on average (Figure 8a). The areal difference in the region exposed to hypoxia with and without P limitation is 21,957 km$^2$, which represents a 34% decrease due to P limitation (Figure 8a). Integrating hypoxia in time and/or space is informative for the effect of P limitation (Figure 9). Typically, the hypoxic area starts

developing in June, extends throughout August to reach its maximum size in early September (Figure 9, top panel). The effect of P limitation on the hypoxic area is large in August and remains significant at the peak extent in September. The decrease in hypoxia extent occurs mainly in zones 2 and 4, with a small additional decrease in zone 3 in September and October (Figure 9, lower panel). In zone 2, the change is
somewhat proportional to the size of the hypoxic area, whereas in zone 4 most of the effect is concentrated in August, hence the large decrease in hypoxic area in August.

## 3.5    Effects of altered nutrient loads

Reducing N and/or P concentration in the Changjiang River affects the spatial distribution and the duration of hypoxia on the shelf (Figure 8b–e). Lowering nutrient concentrations first reduces hypoxia
duration in the northern hypoxia core next to the Changjiang Estuary (Figure 8b–c). The effect of nutrient reduction extends to the southern hypoxia core at higher reduction levels (Figure 8c–d). The spatial differences for each reduction type (N+P, N-only or P-only) are small for a 20% reduction (corresponding to 80% load, Figure 8b) but significant at a 40% nutrient reduction or more (Figure 8c–e). N+P and N-only reductions have a similar effect on hypoxia reduction in the southern area (zone 3) but N+P reduction
enhances hypoxia reduction in the northern area (Zone 2). The N+P and N-only reduction strategies reduce hypoxia to the 2 core zones at a 60% nutrient reduction level (Figure 8d) and lead to normoxic waters with at 80% reduction (Figure 8e). On the contrary, hypoxia remains widely distributed with a P-only strategy, especially in the southern hypoxia core that does not respond to P reduction. Even at 80% P reduction the effect on hypoxia is limited in this area (Figure 8e).

The general outcome of the three strategies is explored by integrating the hypoxic area over time in zones 2-3 and over all areas at various nutrient loadings (Figure 10, Table 2 and supporting Tables S3 and S4). The N+P strategy has the largest effect on hypoxia over the ECS. At 80% and 60% of the original river N+P load, hypoxia is reduced on average by 37% and 69%, respectively (Table 2). The N-only strategy has a lesser effect, whereas the effect of a P-only strategy is limited. The P limitation effect on
hypoxia is not spatially homogeneous and varies between zone 2 (north) and zone 3 (south) (Figure 10, supporting Tables S3 and S4). N+P reduction is clearly the best strategy to reduce hypoxia in zone 2, whereas N-only or P-only have a similar effect at low to intermediate reductions and P-only has less effect

at high P reduction. Although N+P reduction has the largest effect on hypoxia in zone 3, there is little difference with the N-only strategy. In this region the P-only strategy has a much smaller effect on hypoxia.

Overall, the efficiency of a N+P nutrient reduction strategy, i.e., percent reduction in hypoxia (Hbar) per percent reduction in nutrients (Table 2), is 1.87, 1.73, and 1.52 for a 20%, 40% and 60% reduction, respectively. At these levels the average efficiency of a N-only nutrient reduction strategy is 1.31, whereas the average efficiency is only 0.84 for a P-only nutrient reduction strategy. The efficiency is somewhat higher in zone 2 at 2.02, 1.86, and 1.58 for a 20%, 40% and 60% N+P reduction, and an average of 1.31 for N-only and 1.01 for P-only reductions (supporting Table S3). The efficiency is lower in zone 3 at 1.58, 1.55, and 1.44 for a 20%, 40% and 60% N+P reduction, an average of 1.29 for N-only and 0.58 for P-only reductions (supporting Table S4).

## 4    Discussion

### 4.1    Distribution of P limitation on the shelf

Simulated nitrogen and phosphorus distributions off the CE (Figure 3, supporting Figure S1) were in good agreement with observations (Gao et al., 2015; Tseng et al., 2014; Wang et al., 2014). P limitation did not occur in the vicinity of the CE where both $NO_3$ and $PO_4$ are high, as well as along the Jiangsu coast. This northwestern area is N-limited for most of the year, indicating a general lack of influence by Changjiang nutrients. During the productive season, $PO_4$ was depleted rapidly in the plume and the excess nitrogen transported northeastward with the Changjiang plume in late spring and summer.

The resulting P limitation in offshore waters expanded to Jeju Island in the northeast at the seasonal peak of the plume expansion (Figure 5). Observations were limited to the inner shelf and nutrient addition bioassays were not available to compare with the simulated patterns. Nonetheless, early bioassay measurements confirm the presence of P limitation in the region (Harrison et al., 1990). Furthermore, the large-scale distribution of surface nutrients and their limitation effect on primary production are consistent with the main seasonal circulation (Bai et al., 2014; Liu et al., 2021), with observations off the CE (Li et al., 2009; Wang et al., 2015), as well as with observations of Changjiang-related excess $NO_3$ in the

northeastern ECS (Wong et al., 1998) and near Jeju Island (Kodama et al., 2017; Moon et al., 2021). Yet,
additional bioassay data would be useful to validate the simulated patterns of nutrient limitation.

Despite its large-scale distribution, P limitation was most prominent over the southwestern part of the Yangtze Bank (Figure 4) from mid-July to early August (Figure 6), i.e., around the peak of the discharge. The temporal evolution (appearance in late spring, peak in mid-summer, disappearance in late summer and fall) and spatial distribution of P limitation (development downstream in the plume, peak at
mid-shelf, switch to N limitation further offshore) was characteristic of river induced P limitation in open, dispersive systems (Laurent and Fennel, 2017). For instance, in the somewhat similar Mississippi River plume in the northern Gulf of Mexico, seasonal P limitation is observed around mid-shelf at the peak of annual production, between a light limited area in the vicinity of the river and the N-limited downstream/offshore waters (Laurent et al., 2012; Sylvan et al., 2006). Phytoplankton growth limitation
in the ECS followed these general patterns along the CE-JI transect, but over a larger spatial scale given the dispersive nature of the Changjiang plume (Liu et al., 2021).

The particularity of the ECS was the lack of sustained P limitation in the southern hypoxia core (zone 3). Several factors may be at play there: local P regeneration, offshore P supply or the plume orientation. N and P regeneration have different dynamics in the model; regenerated N is partly lost
through denitrification in the sediment, whereas P is regenerated following Redfield (1:16 N:P). "Excess" P (relative to Redfield) is therefore produced during remineralization in the sediment. Yang et al (2017) found highest sediment P recycling and DIP sediment-water flux along the Zhejiang coast (zone 3), which is consistent with the spatial distribution of P limitation in the baseline simulation. P limitation reduces SOC in zones 2 and 4 (Figure 7c) and therefore lowers the "excess" P associated with sediment recycling,
which represents a positive feedback on P limitation in the northeastern area. This decrease is 14% and 16% in zones 2 and 4, respectively (-0.06 and -0.05 mmol P m$^{-2}$ d$^{-1}$), but only 4% in zone 3 (-0.01 mmol P m$^{-2}$ d$^{-1}$) on average. Nonetheless, this effect is somewhat small and therefore P regeneration is only partly responsible for the lack of P limitation in zone 3. Große et al. (2020) recently showed that N supply from the Kuroshio and Taiwan Strait contribute 38% of oxygen consumption in the southern hypoxic
region (equivalent to zone 3). Assuming at least equal contributions of N and P, then offshore supply should be an important factor mitigating P limitation in zone 3. Evidence of P supply from the Kuroshio

(Yang et al., 2012) and from the Taiwan Strait (Huang et al., 2019) to the southern ECS support this mechanism. Those are also a source of $O_2$ to the region (Zhang et al., 2020). Hypoxia occurs later in the southern core region, when the circulation reverses in late summer/early fall. This different timing and the dynamic of the plume at that time (i.e., attached to the coast) may also prevent the development of P limitation in the southern hypoxia core.

## 4.2 Effects on oxygen dynamics and hypoxia

### 4.2.1 Spatial shift and dilution

The hotspot of P limitation off the CE and on the Yangtze Bank had an important controlling effect on primary production, respiration (Figure 6 and Figure 7, supporting Figures S2 and S3) and consequently on the duration and extent of the hypoxic area in summer (Figure 8). In estuarine and coastal systems, P limitation promotes N export to downstream waters, thereby shifting primary production both in time and space (Paerl et al., 2004). This shift is concomitant with a dilution in dispersive systems and therefore limits the deoxygenation of bottom waters (Laurent and Fennel, 2017). The shift/dilution effect occurred in our simulations (Figure 7a). Part of the production was spatially relocated from zones 2 and 4 (spring-summer) to zones 4 and 5 (summer). The total change in primary production in the study area (zones 1-6) varied by less than 1% between the baseline and the NoPlim simulations and therefore the induced variations remained within the shelf. The downstream dilution effect can therefore be quantified by calculating the area-specific change in primary production between the baseline and the NoPlim simulations (supporting Table S5). The annual change in primary production due to P limitation was –3010 mmol $O_2$ m$^{-2}$ in zone 2 and +1310 mmol $O_2$ m$^{-2}$ in zone 5. The decrease in the area-specific value from zone 2 to 5 reveals that primary production was diluted during its relocation downstream. In comparison, the area-specific change in primary production is small in the intermediate zone 4 (–328 mmol $O_2$ m$^{-2}$). The dilution of PP is an important characteristic of P limitation because it modifies POM flux, respiration and therefore the formation of hypoxia (Laurent and Fennel, 2017). Similarly, we can calculate that the annual change in total respiration (WR+SOC) is –1610 mmol $O_2$ m$^{-2}$ in zone 2 and +1020 mmol $O_2$ m$^{-2}$ in zone 5, which represents a 37% decrease of the absolute change (+1230 mmol $O_2$ m$^{-2}$ and a 23.6% decrease for zone 4).

### 4.2.2 Vertical relocation

The change in primary production not only affected total respiration but also its vertical distribution. POM aggregation, and thus sinking rate, is a quadratic function of concentration in the model and therefore lower primary production favors remineralization in the water column rather than in the sediment. This vertical relocation was relatively small in comparison to the dilution effect, the maximum change in the ratio WR:(WR+SOC) was +11%, +16% and –3% in zones 2, 4 and 5, respectively (see supporting Figure

S5). The relative increase in SOC in zone 5 was due to the additional primary production that led to increased deposition. This effect was small in comparison to zones 2 and 4 partly because zone 5 is deeper ($z_{mean}$ = 75 m versus ~37 m).

### 4.2.3 Positive feedback on eutrophication

The net increase in WR relative to SOC is important because of the proportionality between sediment

denitrification and SOC (Fennel et al., 2009). Denitrification ranges from 0.7 (zone 5) to 2.9 (zone 2) mmol N $m^{-2}$ $d^{-1}$ in the baseline simulation (annual average). This is within the lower range of recent observations in the coastal ECS (Lin et al., 2017). The total annual decrease in denitrification resulting from P limitation over the study area (zones 1-6) was 75.0 $x10^8$ mol N (–6.2%). This effect is quite small in comparison to the spatial relocation and dilution of $O_2$ sinks but nevertheless represents a positive

feedback on eutrophication in the ECS.

### 4.2.4 Mitigating effect on the northern hypoxia core and the Yangtze Bank

SOC is an important process in the ECS (Zhou et al., 2017) and the dominant $O_2$ sink below the pycnocline (Zhang et al., 2020). It is therefore not surprising that the change in SOC around the northern hypoxic center and on the Yangtze Bank in July–August, i.e. –11 and –10 mmol $O_2$ $m^{-2}$ $d^{-1}$ (–15% and –20%) in

zone 2 and 4, respectively, had a significant effect on hypoxia (Figure 8a, Figure 9). This effect varies spatially between the two hypoxic centers. Around the northern hypoxic center, the lack of $PO_4$ limits the duration and the spatial expansion of hypoxic conditions and therefore moderates hypoxia. This is where P limitation has the largest effect on hypoxia.

P limitation also prevents the expansion of hypoxic conditions further northwest, on the western side of the Yangtze Bank (Figure 8a). Hypoxia only occurs sporadically in this region and in small patches (909 $km^2$ on average, maximum extent: 3,260 $km^2$) that develop when the plume extends over the bank in July/August. Without P limitation, hypoxia was more frequent and widespread in this area, covering 5,429 $km^2$ on average when hypoxia occurs (maximum: 12,375 $km^2$). The effect on the Yangtze Bank can be related to the lower SOC and a weakening of stratification as the plume mixes with ocean waters downstream. The spatial distribution of the Potential Energy Anomaly (a proxy for stratification, see Große et al. (2020) for details) in summer illustrates this change in stratification over the western Yangtze Bank (supporting Figure S6). This stratification effect is consistent with the idea that hypoxia is tightly controlled by the pycnocline in the northern region (Chi et al., 2017). It was also shown to be an important controlling factor in the context of P limitation in the northern Gulf of Mexico (Laurent and Fennel, 2014). Overall, the area of the shelf that can be exposed to hypoxia decreases by 34% with P limitation, whereas the size of the hypoxic zone in reduced by 48% on average, mainly in August (Figure 8a).

## 4.3    Conceptual model of the effect of P limitation in the ECS

To summarize our findings, we provide a conceptual model of P limitation in the ECS (Figure 11). Along the CE–Jeju Island it follows the general framework of Laurent and Fennel (2017) for multi-dimensional dispersive systems, where the "excess" DIN transported downstream results in a spatial shift and dilution of primary production. The dilution and relocation of $O_2$ sinks are added here, as well as the weakening of stratification downstream. These processes were implied originally. Based on our results, we also add several processes that were not included in the original framework, namely the vertical relocation of $O_2$ sinks and the change in denitrification that represents a small positive feedback on eutrophication. The lack of changes around the southern hypoxic core is specific to the ECS. Große et al. (2020) recently showed with the same model that relative to the Changjiang, nutrient sources from the Taiwan strait and the Kuroshio are important contributors to $O_2$ consumption along the Zhejiang coast (<30°N). This process is added to the conceptual model. This source of nutrients contributes to the occurrence of Harmful Algal Blooms in the area (Che et al., 2022).

As mentioned above, we found similarities in the effect of P limitation between the ECS and other systems. Along the CE-JI line, the dispersive conditions favor the dilution of eutrophication with a positive effect on subsurface water oxygenation within the northern hypoxia core. The same mechanism was found in the northern Gulf of Mexico (Laurent and Fennel, 2014, 2017) and off the Pearl River Estuary (PRE) under severe P limitation (Yu and Gan, 2022). The "dilution effect" (Laurent and Fennel, 2017) is therefore a prevalent consequence of P limitation that is applicable to other river plume systems. We did not find a positive feedback of P limitation on hypoxia as suggested by Yu and Gan (2022) for the PRE. Their study indicates severe, widespread P limitation in offshore waters, which is not in line with our findings for the ECS. This positive feedback may be specific to the PRE system.

## 4.4 Sensitivity of hypoxia to altered nutrient loads

We carried out the first, systematic analysis of the effect of nutrient reduction strategies on hypoxia in the ECS. Große et al. (2020) recently suggested that N management in the Changjiang has the potential to mitigate hypoxia on the shelf. However, they did not assess nutrient reduction strategies. Our results are in line with Große et al. (2020) and further indicate that hypoxia can be eliminated on the shelf at high nutrient reduction levels (Figure 8, Figure 10 and supporting Figure S4). This highlights the direct link between cultural eutrophication and hypoxia in the ECS (Li et al., 2011; Wang et al., 2016). The simulations indicated that management effects vary with the level and type of nutrient reduction, as well as on the area of interest. The N+P nutrient reduction strategy was the most effective to mitigate hypoxia, as reported previously for other coastal systems (Fennel and Laurent, 2018; Kemp et al., 2005; O'Boyle et al., 2015; Paerl, 2009; Scavia and Donnelly, 2007) and for their upstream freshwater systems (Paerl et al., 2016). The dual N+P reduction is generally accepted as the most effective strategy to mitigate eutrophication and hypoxia (Wurtsbaugh et al., 2019). Nonetheless, the efficiency of nutrient reduction strategies (N+P, N-only or P-only) varied between the northern (zone 2) and southern (zone 3) hypoxia cores and was related to the spatial distribution of P limitation. Similar spatial variations in the effectiveness of nutrient reduction strategies (P-only or N-only) were also reported elsewhere (e.g., Chesapeake Bay, Kemp et al. (2005)). In the southern zone 3, where P limitation is not frequent, a N-only strategy was nearly as effective as a N+P strategy in mitigating hypoxia, whereas its efficiency was low

at moderate levels of reductions in the northern zone 2 (Figure 10). This spatially explicit response differs from the northern Gulf of Mexico (Mississippi plume) mentioned above, where nutrient limitation follows an upstream-downstream continuum, as described here along the CE-JI transect.

According to the nutrient load experiments, the level of reduction that is necessary to mitigate hypoxia in the ECS is somewhat moderate due to the high efficiency of the N+P reduction strategy (Figures 8, 10). Long term hypoxia mitigation goals are not in place in the ECS and eutrophication is expected to worsen in the future with the increased use of fertilizer for agriculture (Strokal et al., 2014; Wang et al., 2020). Nonetheless, we explore the feasibility of possible intermediate (-50%) and long term

(-80%) goals to mitigate the hypoxic zone (Figures 9, S4). Our experiments showed that a 50% decrease in the hypoxic area would require a 28% reduction in N+P loads (38% for N-only and 60% for P-only), whereas an 80% decrease in hypoxic area would require a 44% reduction in N+P loads. At such mitigation levels, hypoxia still occurred over a large area in our model but was limited in duration (Figure 8). Since the sediment layer is not explicitly represented in the model (instant remineralization), there is no "legacy"

effect of eutrophication on hypoxia that may be associated with long-term organic matter accumulation in the sediment (Turner et al., 2008; Van Meter et al., 2018). The nutrient reduction levels mentioned above therefore represent a lower limit to the necessary management measures.

In the Changjiang nutrient mitigation experiments we applied the same reduction level to all nutrient subspecies (i.e., dissolved inorganic, dissolved and particulate organic). Changjiang N is

dominated by nitrate, particulate and dissolved organic N contributing only 8–13 % of total N (Große et al., 2020). Most of the mitigation efforts should therefore target the inorganic subspecies. More efficient N recycling from sewage may be a short-term measure to limit DIN load to the CE (Yu et al., 2019). The long-term goal for N reduction may be more challenging due to diffuse sources of DIN in the Changjiang basin (Chen et al., 2019b). Manure and fertilizers are major nutrient sources to the Changjiang and

therefore nutrient reduction targets could partly be met through cost-effective management such as increased manure recycling to cropland  (Strokal et al., 2020; Yu et al., 2019), more efficient fertilizer application (e.g., by increasing farm size, Wu et al., 2018) and, more generally, by increasing nutrient recycling in the Changjiang basin (Yu et al., 2019). Several strategies have been proposed to enhance water quality in the Changjiang (Guo et al., 2021; Yu et al., 2019); our experiments can help estimate

their effectiveness to mitigate hypoxia on the shelf. Further investigations of the link between nutrient limitation in the coastal ECS and the change in land cover and land use in the Changjiang Basin over the last decades may help refine nutrient management strategies.

## 5    Conclusions

We carried out the first systematic assessment of the effects of P limitation and nutrient loadings in the Changjiang-ECS system. P limitation modifies $O_2$ sinks over a large area of the shelf by partly relocating and diluting primary production offshore, away from the locations prone to hypoxia outside the Changjiang Estuary. The resulting horizontal and vertical relocation of $O_2$ sinks drastically reduce the size of the hypoxic area (by about half), mostly off the Changjiang Estuary and on the Yangtze Bank. P limitation had only small effects along the southern Zhejiang coast. The incremental decrease of river nutrients in the Changjiang showed that despite the effect of P limitation on hypoxia, N+P reduction remain the best strategy to mitigate hypoxia. Tentative goals for intermediate (-28%) and long term (-44%) N+P load reductions were proposed to reduce the hypoxic zone by 50% and 80% of its current area, respectively.

*Code and data availability*. The ROMS source code is available from http://myroms.org, last access: 10 August 2019 (Haidvogel et al., 2008). Model results are available on request.

*Supplement*. The supplement related to this article is available online at:

*Authors contributions*. AL and KF conceived the study. HZ set up the model. AL setup the simulations and conducted the analyses. AL wrote the manuscript with input from all co-authors.

*Competing interests*. The authors declare that they have no conflict of interest.

*Acknowledgements*. We thank Compute Canada for providing computing resources under the resource allocation project qqh-593-ac.

*Financial support*. This research has been supported by the Natural Sciences and Engineering Research Council of Canada (NSERC) Discovery program (grant no. RGPIN-2014-03938).

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

**TABLES**

Table 1. Extent of P limitation on the shelf in 2008-2012 and timing of the maximum area.

Table 2. Summary of time-integrated hypoxic area ($km^2$ yr) in the baseline and nutrient reduction experiments. Experiments are indicated as percent load relative to the baseline.

**Table 1.** Extent of P limitation on the shelf in 2008-2012 and timing of the maximum area.

|  | Mean area | Max Area | Integrated area |
| --- | --- | --- | --- |
|  | ($10^3$ km$^2$) | | ($10^3$ km$^2$ yr) |
| 2008 | 34.0 | 71.9 (26-Jul) | 4800 |
| 2009 | 53.4 | 94.6 (04-Jul) | 7454 |
| 2010 | 68.0 | 126.4 (04-Aug) | 9122 |
| 2011 | 31.4 | 86.5 (02-Aug) | 4520 |
| 2012 | 53.7 | 100.0 (16-Jul) | 7580 |
| 2013 | 45.5 | 93.0 (09-Aug) | 6224 |

**Table 2.** Summary of time-integrated hypoxic area (km$^2$ yr) in the baseline and nutrient reduction experiments. Experiments are indicated as percent load relative to the baseline.

| | | Time-integrated hypoxic area (km$^2$ yr) | | | | | |
|---|---|---|---|---|---|---|---|
| | | Baseline | 80% | 60% | 40% | 20% | 0% |
| N+P | 2008 | 582 | 403 | 211 | 66 | 1 | 0 |
| | 2009 | 696 | 439 | 255 | 81 | 1 | 0 |
| | 2010 | 1293 | 870 | 426 | 122 | 0 | 0 |
| | 2011 | 85 | 26 | 5 | 0 | 0 | 0 |
| | 2012 | 440 | 218 | 65 | 7 | 0 | 0 |
| | 2013 | 81 | 31 | 12 | 1 | 0 | 0 |
| | $\Delta\overline{H}$ (%) | **100** | **63** | **31** | **9** | **0** | **0** |
| N-only | 2008 | 582 | 463 | 313 | 143 | 2 | 0 |
| | 2009 | 696 | 524 | 361 | 173 | 1 | 0 |
| | 2010 | 1293 | 1027 | 680 | 216 | 0 | 0 |
| | 2011 | 85 | 33 | 9 | 0 | 0 | 0 |
| | 2012 | 440 | 284 | 140 | 21 | 0 | 0 |
| | 2013 | 81 | 42 | 20 | 6 | 0 | 0 |
| | $\Delta\overline{H}$ (%) | **100** | **75** | **48** | **18** | **0** | **0** |
| P-only | 2008 | 582 | 512 | 428 | 316 | 196 | 93 |
| | 2009 | 696 | 574 | 455 | 346 | 227 | 126 |
| | 2010 | 1293 | 1102 | 905 | 683 | 483 | 302 |
| | 2011 | 85 | 58 | 42 | 27 | 15 | 9 |
| | 2012 | 440 | 339 | 241 | 149 | 79 | 42 |
| | 2013 | 81 | 57 | 40 | 29 | 18 | 8 |
| | $\Delta\overline{H}$ (%) | **100** | **83** | **66** | **49** | **32** | **18** |

# FIGURES

Figure 1. Model domain and bathymetry (left), subregions used for analysis (zones 1-6, upper right) and schematic of the biological model (lower left). Details on $O_2$ sources and sinks are provided in Laurent et al. (2017). On the left panel, the Changjiang is shown in red. The CE-Jeju Island transect is represented by a red line. The black box is the limit of the region shown on the upper right panel.

Figure 2. Changjiang river forcing data. Upper panel: freshwater discharge from the Datong Hydrological station (Figure 1) and nutrient loads. Lower panel: nutrient concentrations and ratio.

Figure 3. a-d: Simulated seasonal means of surface $PO_4$ (2008-2013). The black contours indicate the 25 and 30 isohalines. e-k: Comparison of simulated surface $PO_4$ (colored maps) with the observations of Gao et al. (2015) for the dates indicated in each panel (colored dots). Simulated $PO_4$ corresponds to the mid cruise date.

Figure 4. Annual duration in days of N (left) and P (right) limitation in surface waters. The gray contours indicate the limits of the 6 zones (see Figure 1).

Figure 5. Time series of the nutrient limitation factor along the CE-JI transect (see Figure 1). The color represents the most limiting nutrient at each time/location, either N (blue) or P (red). The thin black line indicates the location of the 29 isohaline. The dots on the right y-axis indicate the limits of zones 2, 4 and 5.

Figure 6. Mean vertical transects along the CE-JI line (see Figure 1) for July 15-31 (a-c) and August 15-31 (d-f). Left panels (a, d): nutrient limitation factors. The color represents the most limiting nutrient at each location/depth, either N (blue) or P (red). Central and right panels: change in primary production (b, e) and water column respiration (c, f) due to P limitation (baseline – NoPlim). Thin grey lines represent the no limitation isoline in panels a, d (limitation factor = 1) and are indicative of PP and WR contours in panels b, e and c, f, respectively.

Figure 7. Spatially integrated change in primary production (a), water column respiration (b) and sediment $O_2$ consumption (c) for zone 2 (blue), zone 4 (orange), zone 5 (red) and zones 1-6 combined (shaded area). Each dot represents a monthly mean for 2008-2013.

Figure 8. Change in hypoxia duration due to (a) P limitation (baseline - NoPlim) and to (b-e) nutrient reduction in the Changjiang. The black contour indicates the area where annual hypoxia duration is at least 1 day in the baseline simulation (2008-2013 mean). The orange contour (panel a) and the dashed, red and blue contours (panels b-e) are the equivalent in the NoPlim and nutrient loads simulations, respectively.

Figure 9. Averaged hypoxic area in the baseline (blue) and NoPlim (red) simulations (top) and difference in time-integrated hypoxic area for each zone (baseline - NoPlim) (bottom). Both panels represent averaged values for 2008-2013.

Figure 10. Effect of nutrient reduction on the size of the hypoxic area (time integrated) for zones 1-6 combined (left), zone 2 (middle) and zone 3 (right). The results presented are an average for 2008-2010. The equivalent for 2011-2013 is presented in supporting Figure S4.

Figure 11. Conceptual model of P limitation on the ECS shelf. Dashed lines indicate the effects without P limitation (NoPlim simulation). 1-5: Conceptual model of river-induced P limitation along the Changjiang-Jeju Island transect adapted from Laurent and Fennel (2017). 1: dominant limitation factor; 2: spatial/temporal relocation of "excess N"; 3: spatial relocation and dilution of PP; 4: spatial shift/dilution and vertical relocation of respiration; 5: weakening of stratification due to plume mixing; 6: mitigating effect on the northern hypoxia core and the Yangtze Bank; 7: no effect of P limitation on the southern hypoxia core; 8: nutrient sources from the Taiwan strait and the Kuroshio as shown by Große et al. (2020); 9: lower denitrification represents a small positive feedback on eutrophication.

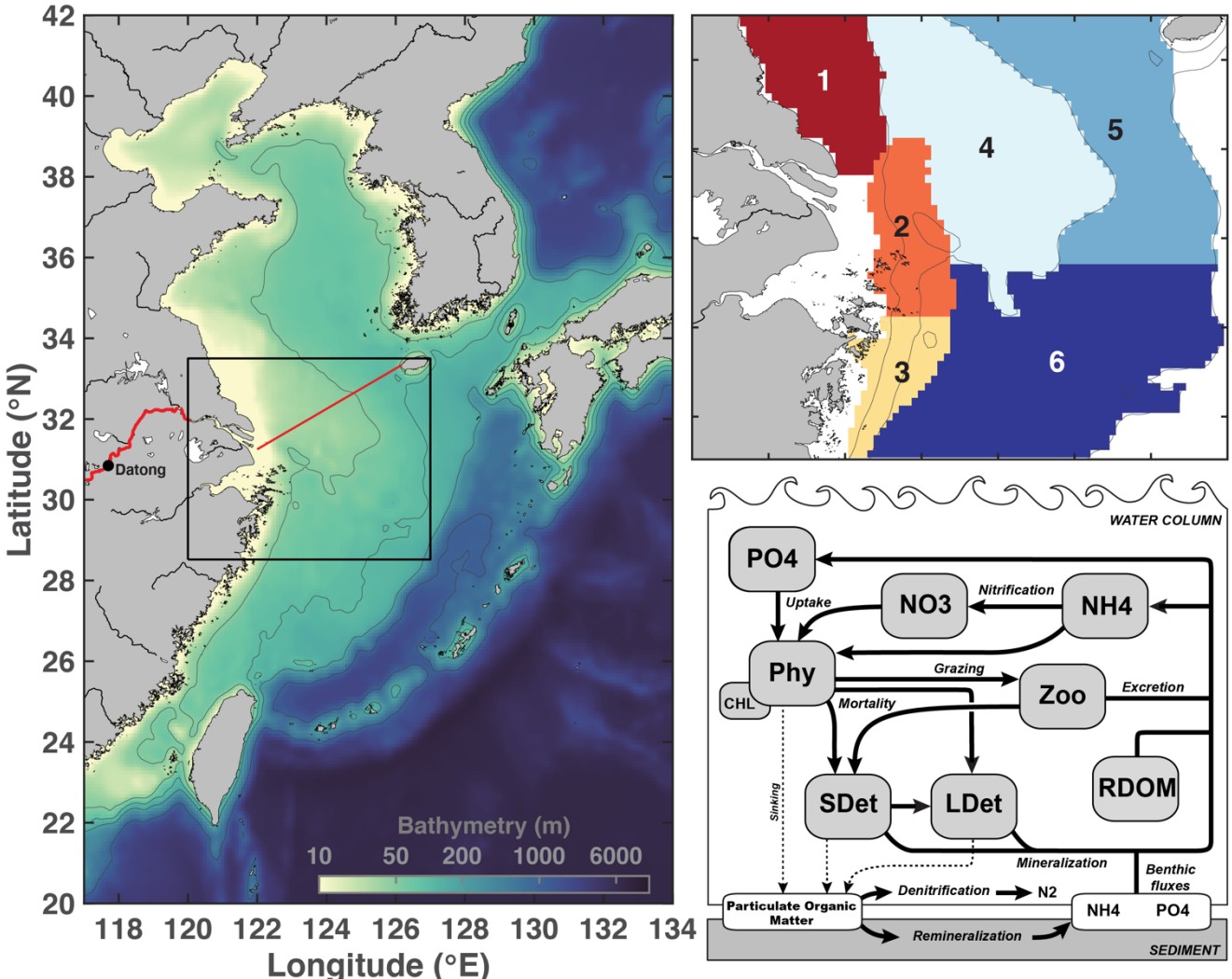

**Figure 1.** Model domain and bathymetry (left), subregions used for analysis (zones 1-6, upper right) and schematic of the biological model (lower right). Details on $O_2$ sources and sinks are provided in Laurent et al. (2017). On the left panel, the Changjiang is shown in red. The CE-Jeju Island transect is represented by a red line. The black box is the limit of the region shown on the upper right panel.

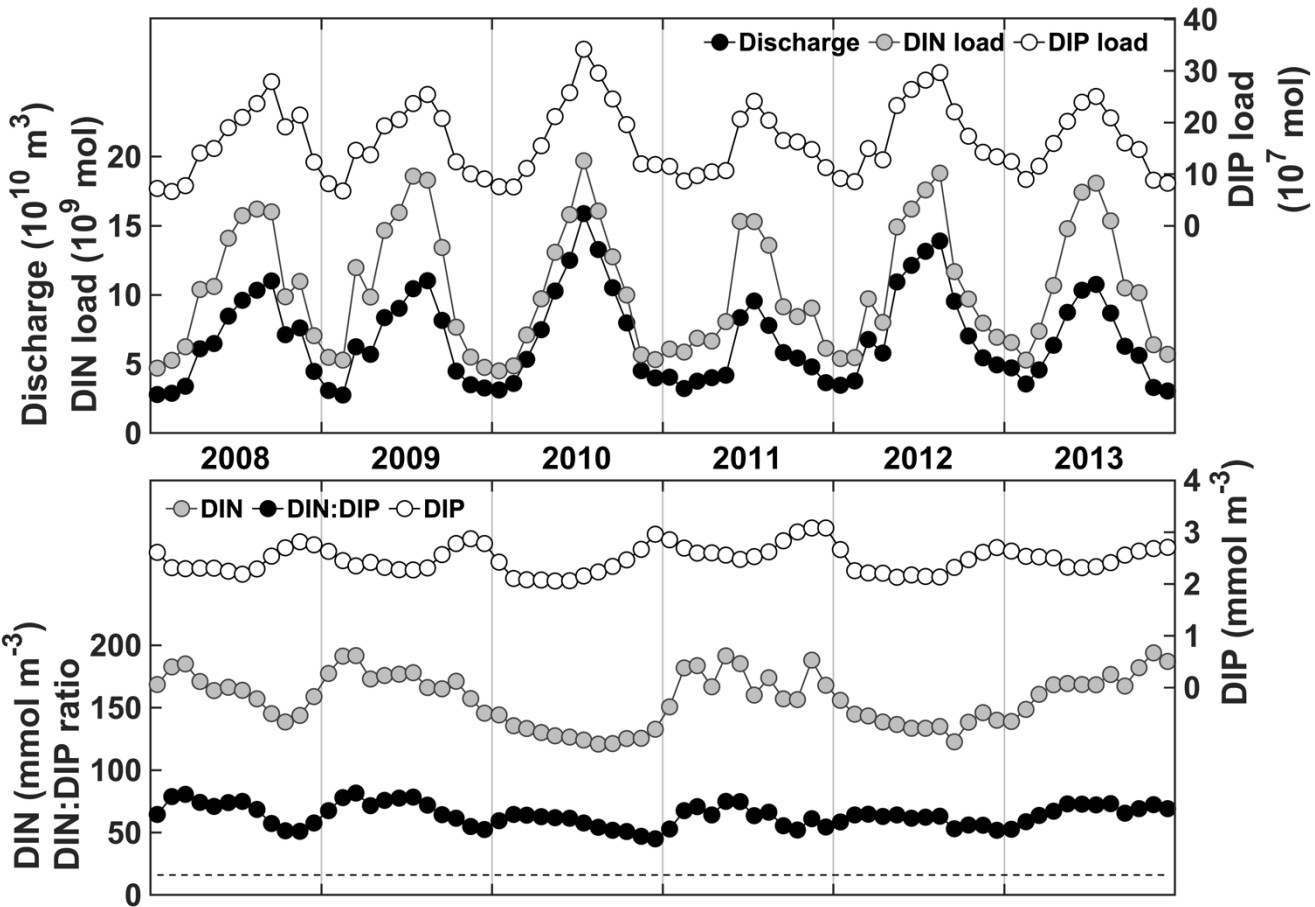

**Figure 2.** Changjiang river forcing data. Upper panel: freshwater discharge from the Datong Hydrological station (Figure 1) and nutrient loads. Lower panel: nutrient concentrations and ratio.

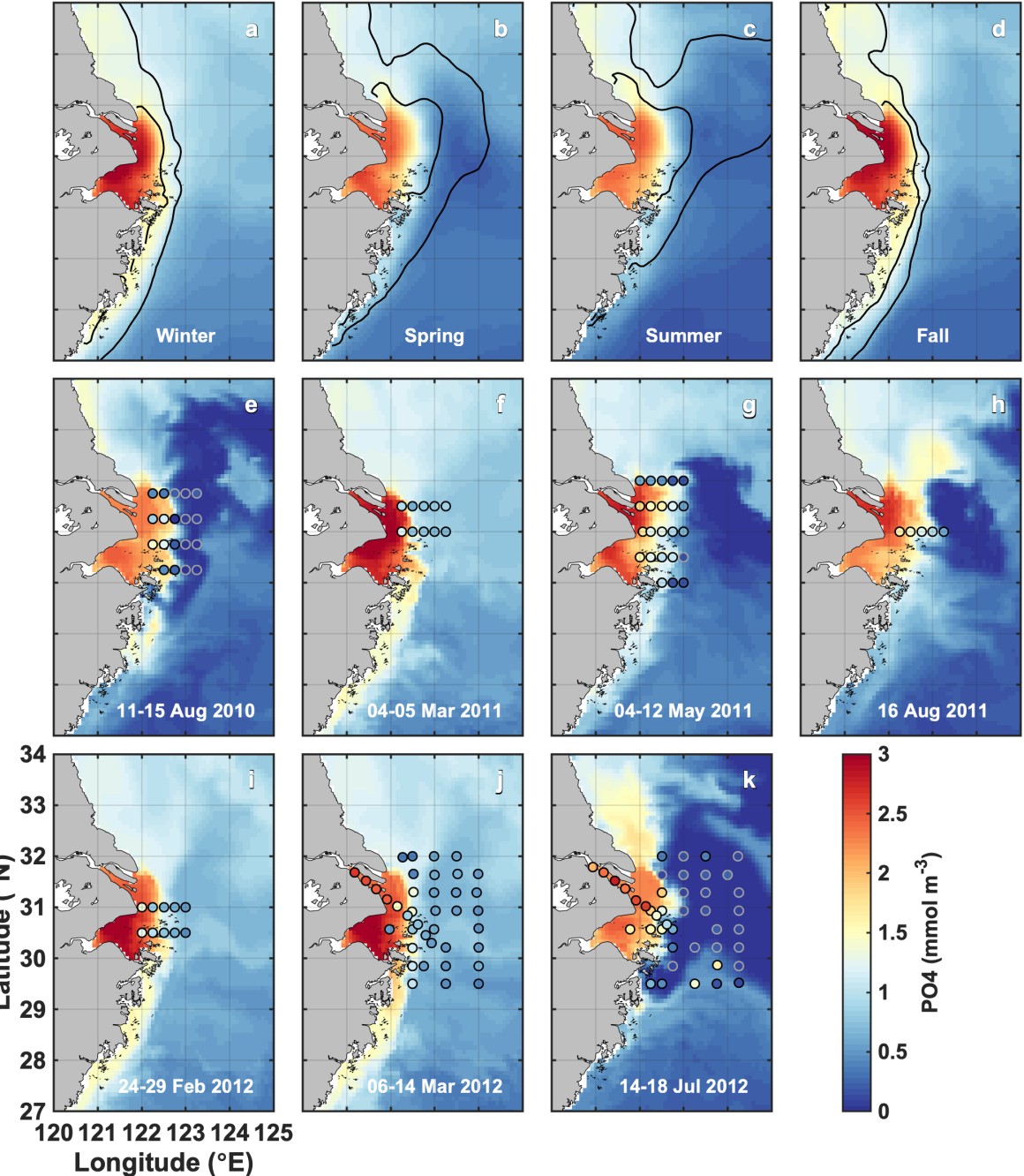

**Figure 3.** a-d: Simulated seasonal means of surface PO$_4$ (2008-2013). The black contours indicate the 25 and 30 isohalines. e-k: Comparison of simulated surface PO$_4$ (colored maps) with the observations of Gao et al. (2015) for the dates indicated in each panel (colored dots). Simulated PO$_4$ corresponds to the average conditions during the cruise.

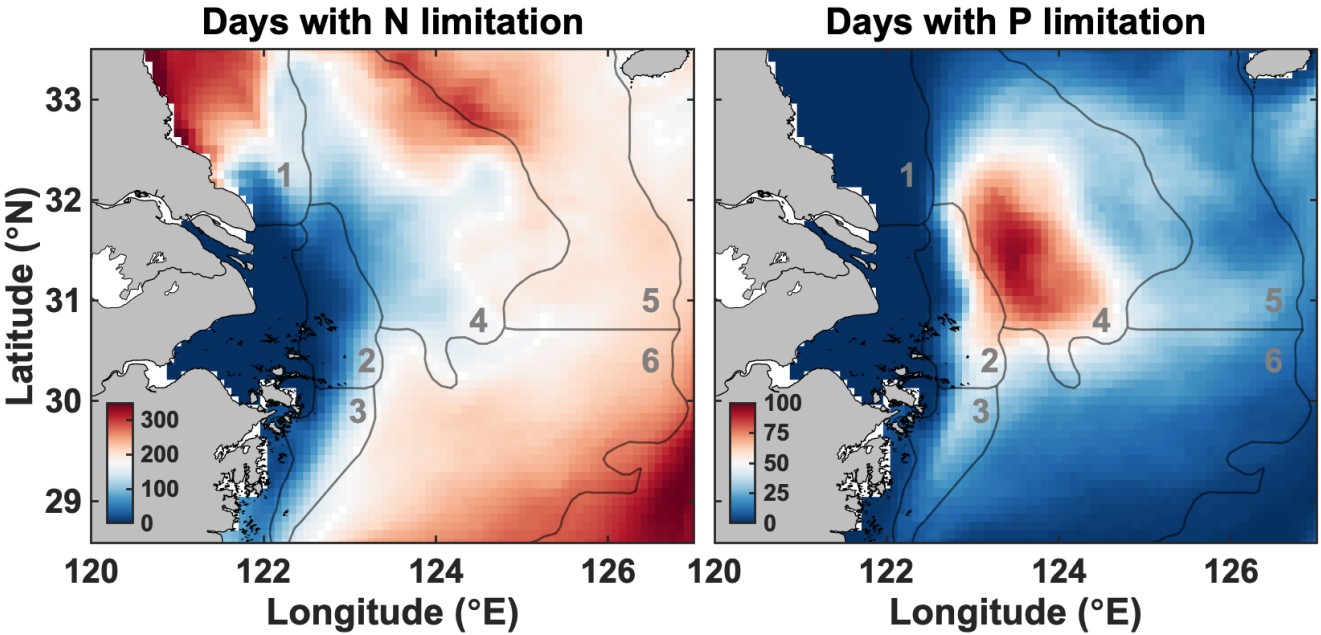

**Figure 4.** Annual duration in days of N (left) and P (right) limitation in surface waters. The gray contours indicate the limits of the 6 zones (see Figure 1).

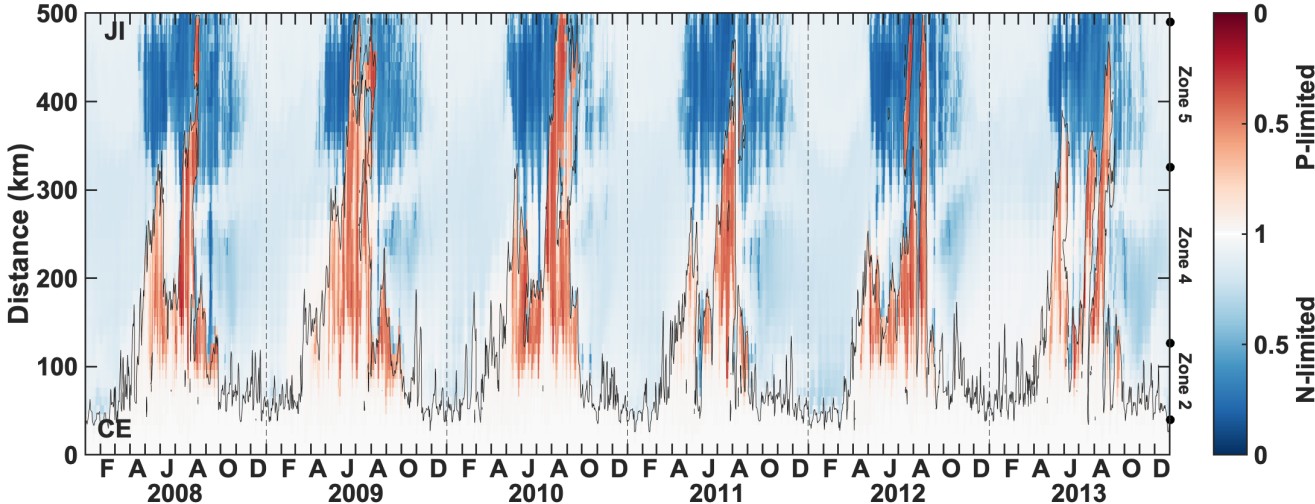

**Figure 5.** Time series of the nutrient limitation factor along the CE-JI transect (see Figure 1). The color represents the most limiting nutrient at each time/location, either N (blue) or P (red). The thin black line indicates the location of the 29 isohaline. The dots on the right y-axis indicate the limits of zones 2, 4 and 5.

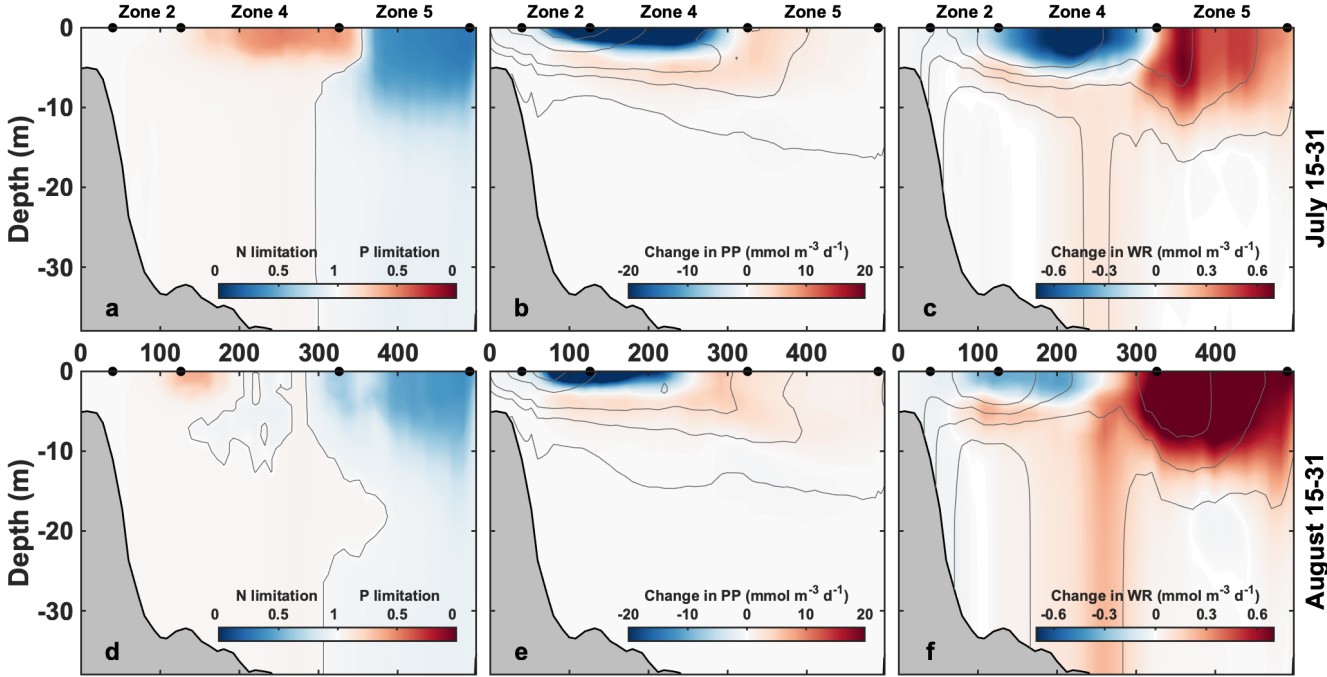

**Figure 6.** Mean vertical transects along the CE-JI line (see Figure 1) for July 15-31 (a-c) and August 15-31 (d-f). Left panels (a, d): nutrient limitation factors. The color represents the most limiting nutrient at each location/depth, either N (blue) or P (red). Central and right panels: change in primary production (b, e) and water column respiration (c, f) due to P limitation (baseline – NoPlim). Thin grey lines represent
the no limitation isoline in panels a, d (limitation factor = 1) and are indicative of PP and WR contours in panels b, e and c, f, respectively. The x-axis is in kilometers from the CE along the CE-JI line.

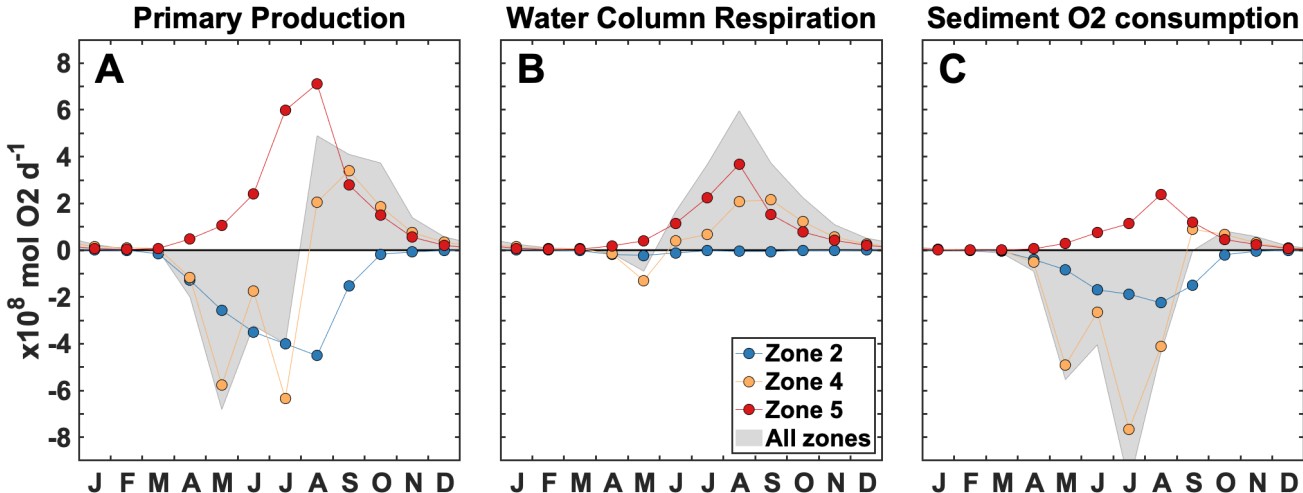

**Figure 7.** Spatially integrated change due P limitation for primary production (a), water column respiration (b) and sediment O₂ consumption (c) in zone 2 (blue), zone 4 (orange), zone 5 (red) and zones 1-6 combined (shaded area). Each dot represents a monthly mean for 2008-2013.

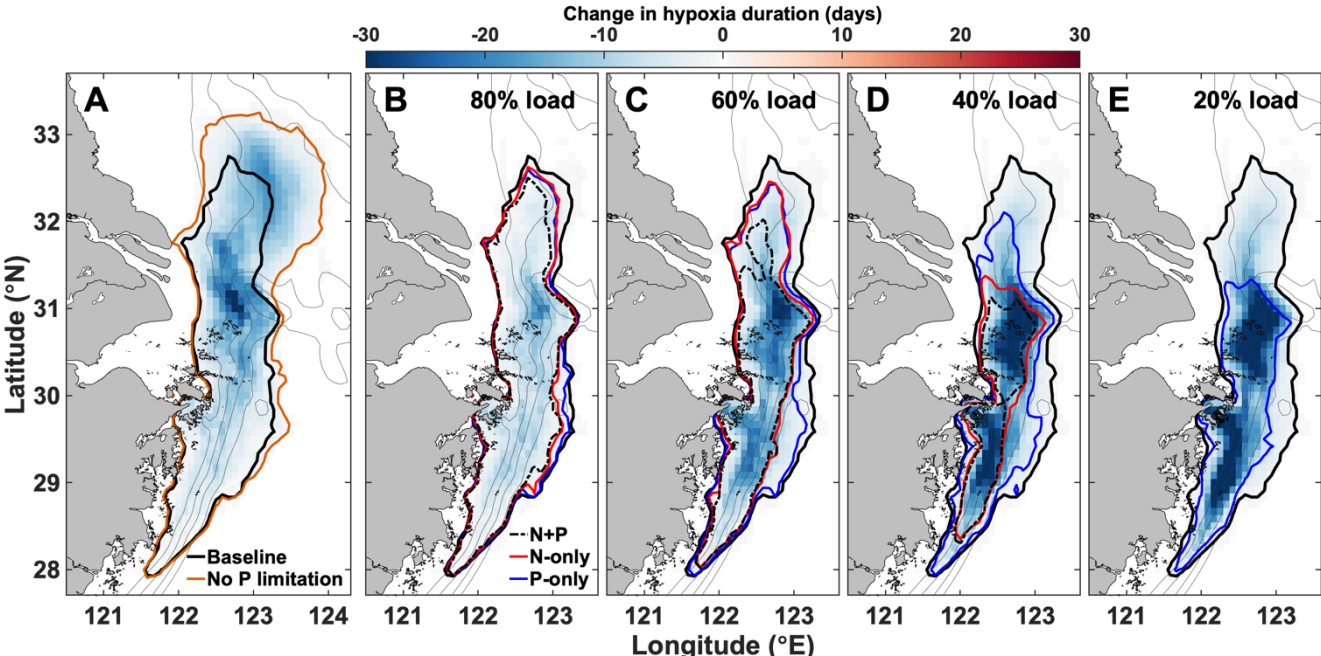

**Figure 8.** Change in hypoxia duration due to (a) P limitation (baseline - NoPlim) and to (b-e) nutrient reduction in the Changjiang. The black contour indicates the area where annual hypoxia duration is at least 1 day in the baseline simulation (2008-2013 mean). The orange contour (panel a) and the dashed, red and blue contours (panels b-e) are the equivalent in the NoPlim and nutrient loads simulations, respectively.

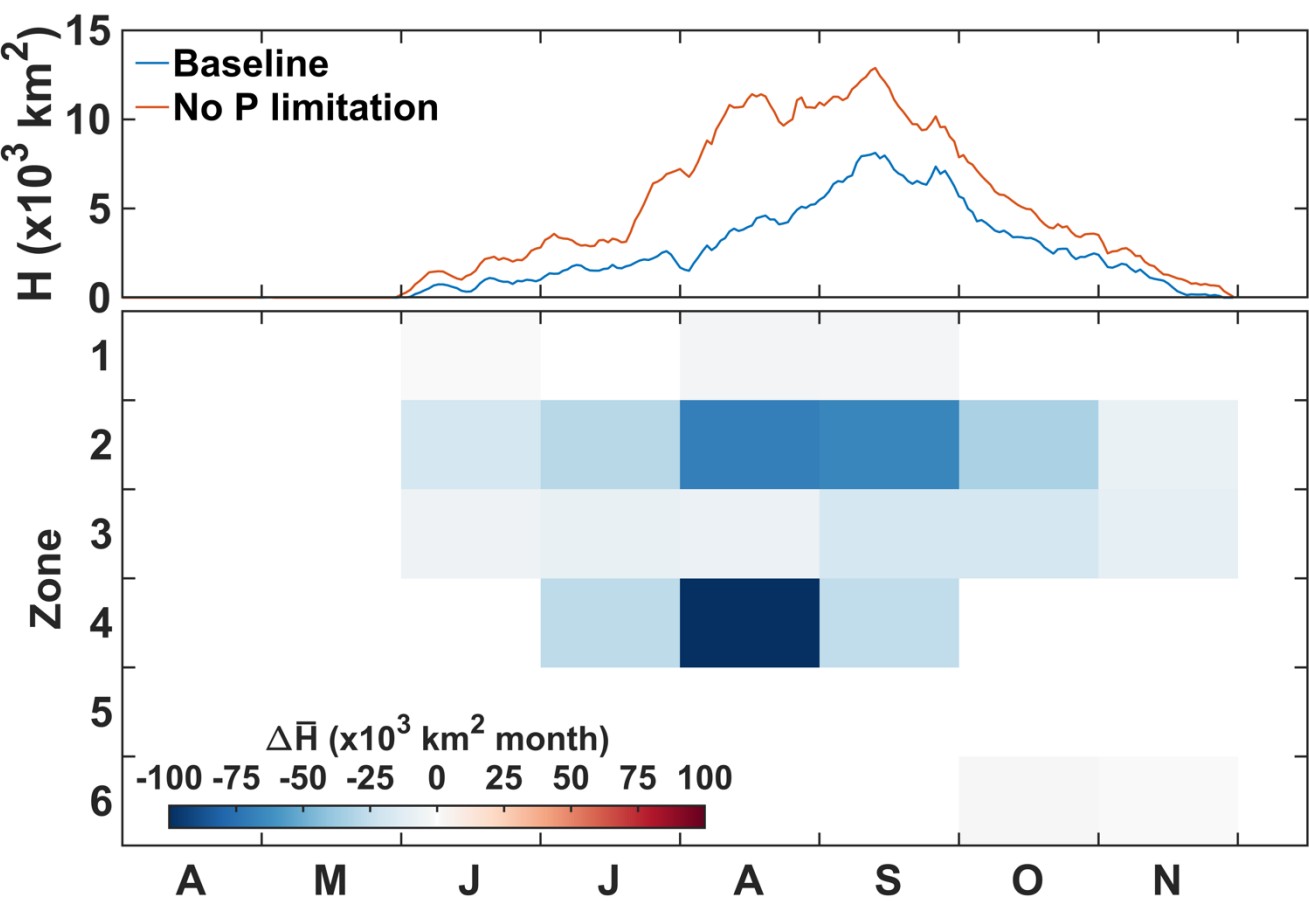

**Figure 9.** Averaged hypoxic area in the baseline (blue) and NoPlim (red) simulations (top) and difference in time-integrated hypoxic area for each zone (baseline - NoPlim) (bottom). Both panels represent averaged values for 2008-2013.

910

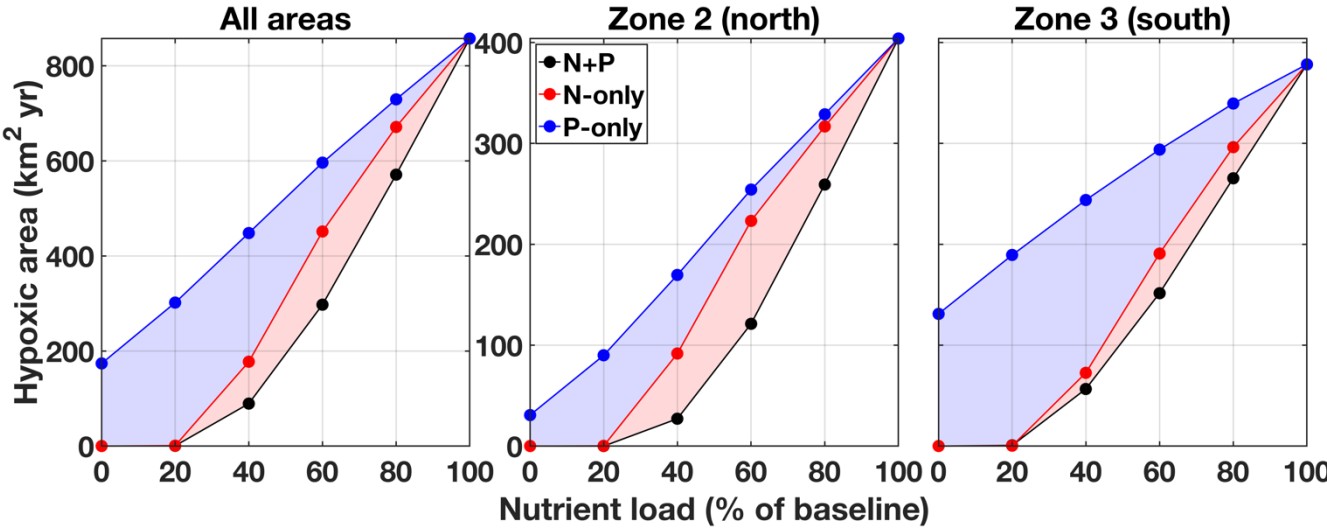

**Figure 10.** Effect of nutrient reduction on the size of the hypoxic area (time integrated) for zones 1-6 combined (left), zone 2 (middle) and zone 3 (right). The results presented are an average for 2008-2010. The equivalent for 2011-2013 is presented in supporting Figure S4.

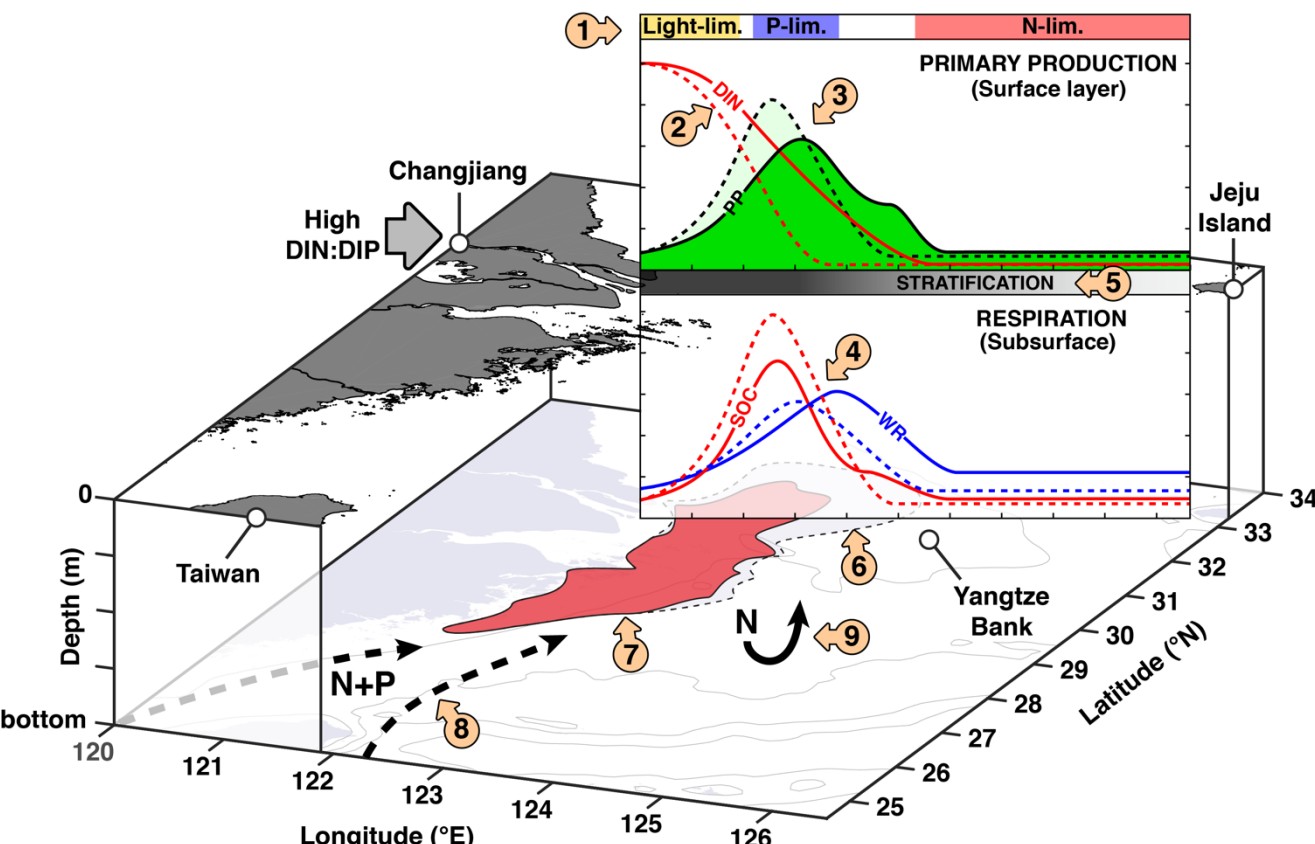

**Figure 11.** Conceptual model of P limitation on the ECS shelf. Dashed lines indicate the effects without P limitation (NoPlim simulation). 1-5: Conceptual model of river-induced P limitation along the Changjiang-Jeju Island transect adapted from Laurent and Fennel (2017). 1: dominant limitation factor; 2: spatial/temporal relocation of "excess N"; 3: spatial relocation and dilution of PP; 4: spatial shift/dilution and vertical relocation of respiration; 5: weakening of stratification due to plume mixing; 6: mitigating effect on the northern hypoxia core and the Yangtze Bank; 7: no effect of P limitation on the southern hypoxia core; 8: nutrient sources from the Taiwan strait and the Kuroshio as shown by Große et al. (2020); 9: lower denitrification represents a small positive feedback on eutrophication.