# Peer review of "Role of phosphorus in the seasonal deoxygenation of the East China Sea shelf"

_Biogeosciences, 2022_

## Author Comment (AC1)

**Detailed responses to reviewer 1** (reviewer comments are included in black, responses in blue font)

**Overview**

Thank you for the opportunity to review the manuscript titled "Role of phosphorus in the seasonal deoxygenation of the East China Sea shelf." I think the manuscript addresses an important problem and has the potential of becoming a worthwhile contribution. The approaches are robust, and the results are elegantly presented. I have offered below a couple of comments, which hopefully can improve the manuscript.

*Response:* We appreciate the constructive feedback. Below we provide a detailed response for each comment.

**General comments**

**Comment:**
1. Introduction: Suggest adding some more background/literature information on the relative importance of P limitation, N limitation, and light limitation in coastal environments, including literature specific to the Changjiang Estuary. This can help the readers set up expectations and better appreciate the results presented in the later sections.

*Response:* Thank you for the suggestion, we will provide additional discussion of the literature in the Introduction of the revised manuscript.

**Comment:**
2. The entire modeling analysis and results are focused on DIN and DIP. What are the roles of other subspecies of N and P? I am not suggesting the authors to re-do the work with TN and TP, but some discussion on the relative importance of DIN (DIP) vs. nondissolved N (P) can be beneficial. This is especially true for the proposed N+P dual nutrient reduction (28% and 44%). A 28% reduction of TN is not equivalent to a 28% reduction of DIN. The same argument is true for P.

*Response:* Chanjiang TN concentration is dominated by nitrate, particulate and dissolved organic N contributing only 8–13 % (Große et al., 2019). Therefore, we focus on DIN (DIP) in our reduction experiments. We will add some discussion about this assumption and the potential effect on our results.

Große, F., Fennel, K. and Laurent, A.: Quantifying the Relative Importance of Riverine and Open-Ocean Nitrogen Sources for Hypoxia Formation in the Northern Gulf of Mexico, J. Geophys. Res. Ocean., 124(8), 5451–5467, doi:10.1029/2019JC015230, 2019.

**Comment:**
3. The analysis of loading reduction scenarios makes the paper stronger and more management relevant. One suggestion is to present the loading reductions in the context of major sources for this system. What are the main contributors of riverine TN and TP loads? How would the proposed reductions be achieved by management actions by

targeting those major sources? Some recommendations based on literature and/or the authors' experience with the watershed will be helpful.

*Response:* We will provide a bit of discussion to put our results in the context of nutrient management.

**Comment:**
4. The majority of the analysis is based on established models. I do not have any issues with the modeling framework, but some statements on the model assumptions and uncertainties are recommended. In addition, have there been any bioassay sampling to verify the model-derived N or limitation status?

*Response:* We will provide some discussion of model assumptions and uncertainties. Unfortunately, we do not have bioassay data to compare with simulated P limitation. Instead, we show that the model reproduces both nitrate and phosphate distribution where cruise data are available.

**Comment:**
5. For the correlation result (Section 3.2), please specify what correlation method was used and justify the choice.

*Response:* We added the term "Pearson's correlations" in sections 3.1 and 3.2.

**Comment:**
6. This research is comprehensive and should have broad relevance. Some discussion that compares the results with other systems or transfer the insights to other systems will be very helpful.

*Response:* We will add a paragraph on the relevance to other systems.

**Comment:**
7. Figure 1: How were the six zones determined. Please clarify in the caption and methods.

*Response:* The following sentence will be added at the end of section 2.2:

"For analysis, the shelf region adjacent to the Changjiang Estuary (CE) is divided into 6 zones. Zone 1 represents the Jiangsu coastal area ($z<25m$, $31.70<lat<33.50$). Zones 2 and 3 are the northern and southern hypoxia cores, respectively, defined in Zhang et al. (2020). Zone 4 represents the Yangtse Bank area ($z<50m$, $30.75<lat<33.50$), whereas Zone 5 ($30.75<lat<33.50$) and 6 ($28.52<lat<30.75$) represent the northern and southern deep shelf waters ($50<z<100m$), respectively."

**Comment:**
8. Please be mindful on the use of significant digits. For example, $68.0 \times 10^3$ km2 (Line 155), $75 \times 10^8$ mol N (line 336), and other occurrences.

*Response:* Done.

---

## Author Comment (AC2)

**Detailed responses to reviewer 2** (reviewer comments are included in black, responses in blue font)

**Overview**

The paper investigated the impact of P limitation on hypoxic zone in the East China Sea using a coupled physical-biogeochemical model. The results shown spatio-temporal variability of P-limitation in this region in details, and provided mechanism explanation on how hypoxic region was modified by N+P, P-only, N-only scenarios from primary production to water column respiration. It also provided management suggestions that N+P reduction was the best strategy to mitigate hypoxia, and quantitative measure on how much hypoxic area would be reduced for intermediate and long-term reduction in N+P reduction. Modelling hypoxia in China coastal ocean is a very hot topic these days. Most of them focused on mechanism explanation. The research had a very special perspective on hypoxia management strategy, which was great and important for hypoxia study in this region. The paper is very well written and organized. It should be published after minor revision.

*Response:* We appreciate the constructive feedback. Below we provide detailed responses for each.

**General comments**

**Comment:**
1. Figure 1: Why zone 1 to 6 was separated like this? Any standard?

*Response:* To clarify our choice of Zones 1-6, the following sentence will be added at the end of section 2.2:

"For analysis, the shelf region adjacent to the Changjiang Estuary (CE) is divided into 6 zones. Zone 1 represents the Jiangsu coastal area (z<25m, 31.70<lat<33.50). Zones 2 and 3 are the northern and southern hypoxia cores, respectively, defined in Zhang et al. (2020). Zone 4 represents the Yangtse Bank area (z<50m, 30.75<lat<33.50), whereas Zone 5 (30.75<lat<33.50) and 6 (28.52<lat<30.75) represent the northern and southern deep shelf waters (50<z<100m), respectively."

**Comment:**
2. In results section 3.3. It mentioned the limitation factor is < 0.85. It should be mentioned in the method section, how this limitation factor is defined and why, although a citation paper has been provided. It will be wonderful to include those crucial information without go back and forth to other references

*Response:* We will add this information in the Methods.

**Comment:**
3. In the results section and Figures, except comparison for the nutrients. A comparison for surface chlorophyll concentration and salinity with satellite data, and hypoxic zone with cruise data overall should also be provided.

*Response:* An extensive model validation was carried out in Zhang et al. (2020) for the same model. We only present nutrient validation here because it is the focus of the investigation and also because PO4 validation was not available in Zhang et al (2020).

**Comment:**
4. Line 270-275: A comparison with the Gulf of Mexico has been mentioned. A recent study on P-limitation on the Pearl River Estuary system. "Reversing impact of phytoplankton phosphorus limitation on coastal hypoxia due to interacting changes in surface production and shoreward bottom oxygen influx" by Yu et al. 2022 (Water research) should also be mentioned. What is the difference and similarity between the three system? Any relation with the land cover and land use change in China? (More urbanization, sewage discharge)?

*Response:* We will provide some discussion of our results in comparison to other systems, such as the Gulf of Mexico and the Pearl River Estuary. The relationship between P limitation, nutrient mitigation and the land cover/land use change in China is an interesting new study and will be mentioned as such in the conclusions.

**Comment:**
5. In the code availability part. Only general ROMS code downloaded was mentioned, did the Fennel module in the most recent ROMS has incorporated all 10 state variables mentioned from Line 94-Line 100. If not, that should be uploaded to Zenodo or somewhere. Also, the parameter scheme in the biological model should be provided. I did not find that in Zhang et al. (2020)

*Response:* Yes, the 10 state variables are available in the current ROMS repository. We used the same parameters as in Zhang et al. (2020), which are available in their Table S1 (supplement). We added this information at the end of section 2.1 as follows:

"The model equations are available in the supporting information of Laurent et al. (2017); setup and validation are described in detail in Zhang et al. (2020). Biogeochemical model parameters are also available in Zhang et al. (Table S1)."

---

## Author Response (AR1)

**Detailed responses to reviewer 1** (reviewer comments are included in black, responses in blue font)

**Overview**

Thank you for the opportunity to review the manuscript titled "Role of phosphorus in the seasonal deoxygenation of the East China Sea shelf." I think the manuscript addresses an important problem and has the potential of becoming a worthwhile contribution. The approaches are robust, and the results are elegantly presented. I have offered below a couple of comments, which hopefully can improve the manuscript.

*Response:* We appreciate the constructive feedback. Below we provide a detailed response for each comment.

**General comments**

**Comment:**
1. Introduction: Suggest adding some more background/literature information on the relative importance of P limitation, N limitation, and light limitation in coastal environments, including literature specific to the Changjiang Estuary. This can help the readers set up expectations and better appreciate the results presented in the later sections.

*Response:* Thank you for the suggestion, we added the following text in the Introduction:

L40-47: *In large eutrophicated estuaries and river plumes (e.g., Mississippi River plume, Changjiang Estuary, Chesapeake Bay) phytoplankton growth is typically limited by light in winter as well as in the highly turbid waters in the vicinity of the river outflow, whereas maximum production often occurs in intermediate waters where both light and nutrients are plentiful (Cloern, 1987; Fennel et al., 2011; Lohrenz et al., 1997, 1999). Further downstream, production is controlled by the availability of nutrients (Fennel and Laurent, 2018; Malone et al., 1996). There, the stoichiometry of river dissolved inorganic N (DIN), P (DIP), and possibly silicate (Si) will partly determine where, when and which nutrient is most limiting (Dagg et al., 2004; Laurent et al., 2012).*

L71-73: *Typically, primary production is limited by light in the Changjiang estuarine and sediment plume waters, whereas nutrient limitation may occur in the transition zone and further offshore (Harrison et al., 1990; Li et al., 2021; Zhu et al., 2009).*

**Comment:**
2. The entire modeling analysis and results are focused on DIN and DIP. What are the roles of other subspecies of N and P? I am not suggesting the authors to re-do the work with TN and TP, but some discussion on the relative importance of DIN (DIP) vs. nondissolved N (P) can be beneficial. This is especially true for the proposed N+P dual nutrient reduction (28% and 44%). A 28% reduction of TN is not equivalent to a 28% reduction of DIN. The same argument is true for P.

*Response:* Changjiang TN concentration is dominated by nitrate, particulate and dissolved organic N contributing only 8–13 % (Große et al., 2019), which is why we

discussed DIN and DIP. However, in the reduction experiments we decrease all N and/or P subspecies. This was not indicated properly in the original manuscript, so we added the following sentence in the methods:

*L149-150: Each reduction level is applied to all nutrient subspecies (i.e., dissolved inorganic, dissolved and particulate organic).*

We also added this statement in the Discussion:

*L463-466: For the Changjiang nutrient mitigation experiments we applied the same reduction level to all nutrient subspecies (i.e., dissolved inorganic, dissolved and particulate organic). Changjiang N is dominated by nitrate, particulate and dissolved organic N contributing only 8–13 % of total N (Große et al., 2020). Most of the mitigation efforts should therefore target the inorganic subspecies.*

Große, F., Fennel, K. and Laurent, A.: Quantifying the Relative Importance of Riverine and Open-Ocean Nitrogen Sources for Hypoxia Formation in the Northern Gulf of Mexico, J. Geophys. Res. Ocean., 124(8), 5451–5467, doi:10.1029/2019JC015230, 2019.

**Comment:**
3. The analysis of loading reduction scenarios makes the paper stronger and more management relevant. One suggestion is to present the loading reductions in the context of major sources for this system. What are the main contributors of riverine TN and TP loads? How would the proposed reductions be achieved by management actions by targeting those major sources? Some recommendations based on literature and/or the authors' experience with the watershed will be helpful.

*Response:* We added more discussion regarding nutrient management at the end of Section 4.4:

*L466-477: More efficient N recycling from sewage may be a short-term measure to limit DIN load to the CE (Yu et al., 2019). The long-term goal for N reduction may be more challenging due to diffuse sources of DIN in the Changjiang basin (Chen et al., 2019b). Manure and fertilizers are major nutrient sources to the Changjiang and therefore nutrient reduction targets could partly be met through cost-effective management such as increased manure recycling to cropland (Strokal et al., 2020; Yu et al., 2019), more efficient fertilizer application (e.g., by increasing farm size, Wu et al., 2018) and, more generally, by increasing nutrient recycling in the Changjiang basin (Yu et al., 2019). Several strategies have been proposed to enhance water quality in the Changjiang (Guo et al., 2021; Yu et al., 2019); our experiments can help estimate their effectiveness to mitigate hypoxia on the shelf. Further investigations on the link between nutrient limitation in the coastal ECS and the change in land cover and land use in the Changjiang Basin over the last decades may help refine nutrient management strategies.*

**Comment:**
4. The majority of the analysis is based on established models. I do not have any issues with the modeling framework, but some statements on the model assumptions and

uncertainties are recommended. In addition, have there been any bioassay sampling to verify the model-derived N or limitation status?

*Response:* We now provide more discussion of model assumptions and uncertainties:

L108-113: *Silicate (Si) is not included in the model as it is assumed to be non-limiting. During the simulation period, Si/N and Si/P were close to and above Redfield, respectively (Shi et al., 2022). The single phytoplankton group includes all phytoplankton types and therefore does not differentiate between diatoms and dinoflagellates. Zhang et al. (2020) showed that these choices are appropriate to represent the bulk surface chlorophyll in the Changjiang plume, as well as spatial and temporal variations in subsurface $O_2$.*

L448-462: *Since the sediment layer is not explicitly represented in the model (instant remineralization), there is no "legacy" effect of eutrophication on hypoxia that may be associated with long-term organic matter accumulation in the sediment (Turner et al., 2008; Van Meter et al., 2018). The nutrient reduction levels mentioned above therefore represent lower limits to the necessary management measures.*

Unfortunately, we do not have bioassay data to compare with simulated P limitation. Instead, we show that the model reproduces both nitrate and phosphate distribution where cruise data are available (L310-L314). We added the following statements regarding the lack of bioassay data:

L308-310: *Observations were limited to the inner shelf and nutrient addition bioassays were not available to compare with the simulated patterns. Nonetheless, early bioassay measurements confirm the presence of P limitation in the region (Harrison et al., 1990).*

L314-315: *Yet, additional bioassay data would be useful to validate the simulated patterns of nutrient limitation.*

**Comment:**
5. For the correlation result (Section 3.2), please specify what correlation method was used and justify the choice.

*Response:* We added the term "Pearson's correlations" in sections 3.1 and 3.2.

**Comment:**
6. This research is comprehensive and should have broad relevance. Some discussion that compares the results with other systems or transfer the insights to other systems will be very helpful.

*Response:* We added the following text for comparison with other systems (NGOM, PRE):

L420-428: *As mentioned above, we found similarities in the effect of P limitation between the ECS and other systems. Along the CE-JI line, the dispersive conditions favor the*

*dilution of eutrophication with a positive effect on subsurface water oxygenation within the northern hypoxia core. The same mechanism was found in the northern Gulf of Mexico (Laurent and Fennel, 2014, 2017) and off the Pearl River Estuary (PRE) under severe P limitation (Yu and Gan, 2022). The "dilution effect" (Laurent and Fennel, 2017) is therefore a prevalent consequence of P limitation that is applicable to other river plume systems. We did not find a positive feedback of P limitation on hypoxia as suggested by Yu and Gan (2022) for the PRE. Their study indicates severe, widespread P limitation in offshore waters, which is not in line with our findings for the ECS. This positive feedback may be specific to the PRE system.*

**Comment:**
7. Figure 1: How were the six zones determined. Please clarify in the caption and methods.

*Response:* The following sentence was added at the end of section 2.3:

L151-155: *For analysis, the shelf region adjacent to the Changjiang Estuary (CE) is divided into 6 zones. Zone 1 represents the Jiangsu coastal area (z<25m, 31.70<lat<33.50). Zones 2 and 3 are the northern and southern hypoxia cores, respectively, defined in Zhang et al. (2020). Zones 4 represents the Yangtse Bank area (z<50m, 30.75<lat<33.50), whereas Zones 5 (30.75<lat<33.50) and 6 (28.52<lat<30.75) represent the northern and southern deep shelf waters (50<z<100m), respectively.*

**Comment:**
8. Please be mindful on the use of significant digits. For example, 68.0 ×103 km2 (Line 155), 75 x108 mol N (line 336), and other occurrences.

*Response:* Done.

**Detailed responses to reviewer 2** (reviewer comments are included in black, responses in blue font)

**Overview**

The paper investigated the impact of P limitation on hypoxic zone in the East China Sea using a coupled physical-biogeochemical model. The results shown spatio-temporal variability of P-limitation in this region in details, and provided mechanism explanation on how hypoxic region was modified by N+P, P-only, N-only scenarios from primary production to water column respiration. It also provided management suggestions that N+P reduction was the best strategy to mitigate hypoxia, and quantitative measure on how much hypoxic area would be reduced for intermediate and long-term reduction in N+P reduction. Modelling hypoxia in China coastal ocean is a very hot topic these days. Most of them focused on mechanism explanation. The research had a very special perspective on hypoxia management strategy, which was great and important for hypoxia study in this region. The paper is very well written and organized. It should be published after minor revision.

*Response:* We appreciate the constructive feedback. Below we provide detailed responses for each.

**General comments**

**Comment:**
1. Figure 1: Why zone 1 to 6 was separated like this? Any standard?

*Response:* To clarify our choice of Zones 1-6, the following sentence was added at the end of section 2.3:

L151-155: *For analysis, the shelf region adjacent to the Changjiang Estuary (CE) is divided into 6 zones. Zone 1 represents the Jiangsu coastal area (z<25m, 31.70<lat<33.50). Zones 2 and 3 are the northern and southern hypoxia cores, respectively, defined in Zhang et al. (2020). Zones 4 represents the Yangtse Bank area (z<50m, 30.75<lat<33.50), whereas Zones 5 (30.75<lat<33.50) and 6 (28.52<lat<30.75) represent the northern and southern deep shelf waters (50<z<100m), respectively.*

**Comment:**
2. In results section 3.3. It mentioned the limitation factor is < 0.85. It should be mentioned in the method section, how this limitation factor is defined and why, although a citation paper has been provided. It will be wonderful to include those crucial information without go back and forth to other references

*Response:* We added a new section describing nutrient limitation in the model (Section 2.2, L130-139).

**Comment:**

3. In the results section and Figures, except comparison for the nutrients. A comparison for surface chlorophyll concentration and salinity with satellite data, and hypoxic zone with cruise data overall should also be provided.

*Response:* An extensive model validation was carried out in Zhang et al. (2020) for the same model. We only present nutrient validation here because it is the focus of the investigation and also because $PO_4$ validation was not available in Zhang et al (2020). We added the following statement in the Methods:

L111-113: *Zhang et al. (2020) showed that these choices are appropriate to represent the bulk surface chlorophyll in the Changjiang plume, as well as spatial and temporal variations in subsurface $O_2$.*

**Comment:**

4. Line 270-275: A comparison with the Gulf of Mexico has been mentioned. A recent study on P-limitation on the Pearl River Estuary system. "Reversing impact of phytoplankton phosphorus limitation on coastal hypoxia due to interacting changes in surface production and shoreward bottom oxygen influx" by Yu et al. 2022 (Water research) should also be mentioned. What is the difference and similarity between the three system? Any relation with the land cover and land use change in China? (More urbanization, sewage discharge)?

*Response:* We added the following text for comparison with the northern Gulf of Mexico and the Pearl River Estuary:

L420-428: *As mentioned above, we found similarities in the effect of P limitation between the ECS and other systems. Along the CE-JI line, the dispersive conditions favor the dilution of eutrophication with a positive effect on subsurface water oxygenation within the northern hypoxia core. The same mechanism was found in the northern Gulf of Mexico (Laurent and Fennel, 2014, 2017) and off the Pearl River Estuary (PRE) under severe P limitation (Yu and Gan, 2022). The "dilution effect" (Laurent and Fennel, 2017) is therefore a prevalent consequence of P limitation that is applicable to other river plume systems. We did not find a positive feedback of P limitation on hypoxia as suggested by Yu and Gan (2022) for the PRE. Their study indicates severe, widespread P limitation in offshore waters, which is not in line with our findings for the ECS. This positive feedback may be specific to the PRE system.*

The relationship between P limitation, nutrient mitigation and the land cover/land use change in China is an interesting new study and was mentioned at the end of the Discussion:

L475-477: *Further investigations on the link between nutrient limitation in the coastal ECS and the change in land cover and land use in the Changjiang Basin over the last decades may help refine nutrient management strategies.*

**Comment:**

5. In the code availability part. Only general ROMS code downloaded was mentioned, did the Fennel module in the most recent ROMS has incorporated all 10 state variables mentioned from Line 94-Line 100. If not, that should be uploaded to Zenodo or somewhere. Also, the parameter scheme in the biological model should be provided. I did not find that in Zhang et al. (2020)

*Response:* Yes, the 10 state variables are available in the current ROMS repository. We used the same parameters as in Zhang et al. (2020), which are available in their Table S1 (supplement). We added this information at the end of section 2.1 as follows:

L126-129: *The model equations are available in the supporting information of Laurent et al. (2017); setup and validation are described in detail in Zhang et al. (2020). Biogeochemical model parameters are also available in Zhang et al. (Table S1).*

**Detailed responses to reviewer 3** (reviewer comments are included in black, responses in blue font)

**Overview**

The manuscript presents a modelling study of the effects of nutrient limitation on deoxygenation in the East China Sea shelf. The manuscript is well composed, and results are clearly presented using meaningful figures. In particular, the conceptual scheme presented in Fig. 11 well resume the main results of the manuscript. Thus, my suggestion is to publish the manuscript after minor revisions (listed hereafter).

**Comment:**
1. L75. Consider changing "simulations with and without P" in "simulations with and without P limitation".

*Response:* Done.

**Comment:**
2. L79. I suggest specifying that the scenario simulations are all made with reduction of nutrient loads.

*Response:* We modified the sentence as follows:

L91-92: *… by conducting scenario simulations with reduced nutrient loads from the Changjiang.*

**Comment:**
3.  L91-102. Even if references are provided, maybe some additional details on the simulation would be helpful for the reader. For instance, are the circulation and the biogeochemical model coupled online or offline? Since the simulation results can be in principle affected by any nutrient inputs, information about treatment of nutrient fluxes at boundaries and at surface could be of interest for the reader.

*Response:*  The detailed model description is available in Zhang et al (2020) but we added additional relevant information to the Method section:

L103-104: *The circulation model is coupled online with the biogeochemical model of Fennel et al. (2006, 2011)*

L118-125: *The model includes seven rivers. Daily freshwater discharge observations from Datong Hydrological Station (**Error! Reference source not found.**) and monthly $NO_3$ and $PO_4$ loads from the Global NEWs model (Wang et al., 2015) are used to represent Changjiang conditions. Monthly or annual climatologies are used for the other rivers. Boundary conditions for temperature, salinity, $NO_3$, $PO_4$ and $O_2$ are specified from the World Ocean Atlas (Garcia et al., 2013a,b; Locarnini et al., 2013; Zweng et al., 2013), whereas horizontal velocities and sea surface elevation use the SODA data set (Carton and Giese, 2008). A small positive value is used for all other biological*

*variables. Surface forcing is prescribed using the ECMWF ERA-Interim dataset (Dee et al., 2011).*

We also added a new section describing nutrient limitation in the model (Section 2.2, L130-139).

**Comment:**
4. L106-108. My suggestion is to somehow revert how the noPlim simulation is introduced by firstly clarifying that the objective is to have a simulation without P limitation, and then by specifying that this has been implemented disabling P in the biogeochemical model.

*Response:* We modified the text as follows:

L142-145: *The "baseline" simulation, as described above, is identical to Zhang et al. (2020). To assess the effect P limitation, the baseline is compared to a simulation without P limitation ("noPlim"), that is implemented by disabling P in the biological model but otherwise keeping everything identical to the "baseline" simulation.*

**Comment:**
5. L110. Maybe the final sentence of the paragraph ("Otherwise […] simulation") can be removed if the previous suggestion 4 will be accepted by the Authors. Sub-section 3.1 is inserted in the Results section. However, since riverine input is a forcing of the simulations, I am wondering if it would be more appropriate to move 3.1 in section 2 (Methods).

*Response:* See response to Comment 4 above. Regarding section 3.1, we provide some high-level analysis of the river forcing and therefore feel that it is more appropriate to keep it in the Results section.

**Comment:**
6. L130-135. Validation is made considering observations limited to the surface layer. I recognize that validation can be made only with available data and that the main aim of the manuscript is not the validation of the model, however the Authors could introduce in the Discussion some comments about the implications of the use of a relatively limited dataset or the expected impact of additional observations.

*Response:* We added some discussion of the model uncertainties as follows:

L108-113: *Silicate (Si) is not included in the model as it is assumed to be non-limiting. During the simulation period, Si/N and Si/P were close to and above Redfield, respectively (Shi et al., 2022). The single phytoplankton group includes all phytoplankton types and therefore does not differentiate between diatoms and dinoflagellates. Zhang et al. (2020) showed that these choices are appropriate to represent the bulk surface chlorophyll in the Changjiang plume, as well as spatial and temporal variations in subsurface $O_2$.*

*L448-462: Since the sediment layer is not explicitly represented in the model (instant remineralization), there is no "legacy" effect of eutrophication on hypoxia that may be associated with long-term organic matter accumulation in the sediment (Turner et al., 2008; Van Meter et al., 2018). The nutrient reduction levels mentioned above therefore represent lower limits to the necessary management measures.*

Regarding the limited set of observations, we added this statement at the beginning of the Discussion:

*L308-315: Observations were limited to the inner shelf and nutrient addition bioassays were not available to compare with the simulated patterns. Nonetheless, early bioassay measurements confirm the presence of P limitation in the region (Harrison et al., 1990). Furthermore, the large-scale distribution of surface nutrients and their limitation effect on primary production are consistent with the main seasonal circulation (Bai et al., 2014; Liu et al., 2021), with observations off the CE (Li et al., 2009; Wang et al., 2015), as well as with observations of Changjiang-related excess $NO_3$ in the northeastern ECS (Wong et al., 1998) and near Jeju Island (Kodama et al., 2017; Moon et al., 2021). Yet, additional bioassay data would be useful to validate the simulated patterns of nutrient limitation.*

**Comment:**
7. Figure S1. If I am not wrong this figure shows the same results of Fig. 2 of Zhang et al. 2020. I think that it should be somehow highlighted that this material has been already published elsewhere.

*Response:* Figure S1 is similar to Figure 2 in Zhang et al (2020) except that here we plot the average simulated surface conditions over the cruise time range whereas Zhang et al (2020) use mid cruise condition so the two figures are somewhat different. However, we noticed that there was an error in the captions of Figure 3 and S1, which was corrected as follows:

"Simulated XX corresponds to the average conditions during the cruise."

where XX is either NO3 (Figure S1) or PO4 (Figure 3).

**Comment:**
8. L191-192. Since only regions 2, 4 and 5 are shown in Fig. 7, I suggest removing the "e.g." used into the brackets: "more concentrated in the smallest area (zone 2) and more diffuse in the larger areas (zones 4 and 5)."

*Response:* Done.

**Comment:**
9. L75. Consider changing "simulations with and without P" in "simulations with and without P limitation".

*Response:* Done.

**Comment:**
10.  L231. Maybe it would be clearer "The N+P and N-only reduction strategies" instead of "The 2 strategies".

*Response:*  Done.

**Comment:**
11. Figure 6. Even if one can guess that the increasing numbers on the horizontal axis are the distance along the CE-JI line, it should be better to specify this aspect in the figure or in the caption.

*Response:* This information was added in the caption as follows:

"The x-axis is in kilometers from the CE along the CE-JI line".

**Comment:**
12. Figure 7. I think that it could be useful to specify that the change is related to P limitation.

*Response:* Done.